# Energy-level quantization and single-photon control of phase slips in YBa$_2$Cu$_3$O$_{7-x}$ nanowires

M. Lyatti [1,2,3]*, M.A. Wolff[1], I. Gundareva[2,3], M. Kruth[4], S. Ferrari[1], R.E. Dunin-Borkowski [3] & C. Schuck[1]

Significant progress has been made in superconducting quantum circuits. However new quantum devices that have longer decoherence times at higher temperatures are urgently required for quantum technologies. Superconducting nanowires with quantum phase slips are promising candidates for use in novel quantum devices. Here, we demonstrate YBa$_2$Cu$_3$O$_{7-x}$ nanowires with phase-slip dynamics and study their switching-current statistics at temperatures below 20 K. We apply theoretical models developed for Josephson junctions and show that our results provide strong evidence for energy-level quantization in the nanowires. The crossover temperature to the quantum regime of 12–13 K and the lifetime in the excited state exceeding 20 ms at 5.4 K are superior to those in conventional Josephson junctions. We also show how the absorption of a single photon changes the phase-slip and quantum state of a nanowire, which is important for the development of single-photon detectors with high operating temperature and superior temporal resolution.

[1] Institute of Physics, University of Münster, 48149 Münster, Germany. [2] Kotelnikov IRE RAS, 125009 Moscow, Russia. [3] PGI-5, Forschungszentrum Jülich, 52425 Jülich, Germany. [4] ER-C 2, Forschungszentrum Jülich, 52425 Jülich, Germany. *email: matvey_l@mail.ru

Superconducting quantum circuits are based on electrical inductor-capacitor (LC) oscillators, in which the Josephson effect contributes the nonlinearity that is required for selective access to quantum levels[1]. Historically, tunnel Josephson junctions (JJs) have played a major role in studies of macroscopic quantum phenomena and, nowadays, most superconducting quantum circuits are based on them. However, the Josephson effect also occurs in structures with a non-tunneling conductivity, which together with tunnel JJs form a class of superconducting weak links[2]. Nanowires with quantum phase slips caused by fluctuations of the order parameter are particularly interesting superconducting weak links with direct conductivity, as they can be used in superconducting quantum circuits[3–6]. The potential of these nanowires lies in their long-lived excited states, which result from their low sensitivity to charge noise and critical current noise[3]. The electrodynamics of superconducting nanowires with strong fluctuations of the order parameter are not well understood, but are likely to be governed by principles similar to those for single JJs. Notably, it is possible to describe both systems by the resistively and capacitively shunted junction (RCSJ) model[7–9], when applying minor modifications that account for the differences between phase-slip nanowires (PSNs) and JJs (see Supplementary Note 1). Here, we use the term PSN to refer to a superconducting nanowire that has a finite critical current and the resistive state occurring due to phase slippage.

A tunnel JJ exhibits two macroscopic quantum phenomena that are important for quantum circuits: macroscopic quantum tunneling (MQT) and energy-level quantization (ELQ) that were experimentally observed for JJs from low-temperature (low-$T_c$) and high-temperature (high-$T_c$) cuprate superconductors[10–15]. MQT has been demonstrated for PSNs fabricated from different low-$T_c$ superconductors[16–18]. However, there is a significant difference between a JJ and a low-$T_c$ PSN, because the physical mechanisms that determine the frequency of plasma oscillations have a very different nature. The zero-bias plasma frequency of a JJ $\omega_{p0} = (2eI_c/C\hbar)^{1/2}$ (referred to as a Josephson plasma frequency) is given by the resonant frequency of an $L_J C$ circuit consisting of a Josephson inductance $L_J = \hbar/2eI_c\cos\varphi$ and a junction capacitance $C$[19]. Here, $I_c$ is critical current, $e$ is the electron charge, $\hbar = h/2\pi$ is reduced Planck's constant, and $\varphi$ is phase difference across a JJ. When compared with tunnel JJs, the nanowires typically have very small intrinsic capacitances. In a low-$T_c$ nanowire, the energy of Josephson plasma oscillations is much higher than the superconducting energy gap $2\Delta$, which makes such oscillations impossible, as shown in Fig. 1a. In a pioneering study[16], Giordano proposed that the plasma frequency of low-$T_c$ PSNs is limited by another physical mechanism and scales with $\Delta$. This was experimentally confirmed by measurements of the crossover temperature between MQT and thermal activation (TA) escape mechanisms for different low-$T_c$ PSNs[17,18]. As a result of the very high plasma frequency, it is therefore unlikely that more than one energy level exists in a low-$T_c$ PSN.

In contrast, in high-$T_c$ superconductors the superconducting energy gap is much larger and Josephson plasma oscillations are allowed, as shown in Fig. 1a. In addition to having a large superconducting energy gap of $\Delta = 25$–30 meV[20], YBa$_2$Cu$_3$O$_{7−x}$ (YBCO) nanowires demonstrate a single-valued sine-like current-phase relation, even at temperatures close to zero[21].

Here, we show that YBCO nanowires are promising candidates for realizing superconducting quantum circuits. Our measurements of switching-current statistics for ultra-thin YBCO nanowires with phase-slip dynamics provide clear evidence of ELQ in the nanowires.

## Results

**Energy-level quantization.** We performed electrical transport measurements on 2-μm-long 8.2-nm-thick YBCO nanowires with nominal widths <300 nm on a (100) SrTiO$_3$ (STO) substrate. Figure 1b shows a scanning electron micrograph of a representative nanowire patterned using focused ion ion beam (FIB) milling across a 10-μm-wide microbridge. All of the nanowires showed current–voltage (IV) curves that were characteristic of phase slippage, as described in previous work[22]. Based on the linear dependence of the critical current and normal-state resistance on nominal nanowire width and thickness, we determined the effective nanowire width and thickness that we use below rather than the nominal ones. Significantly, the switching-current statistics of nanowires with effective widths of <100 nm cannot be explained using models developed for low-$T_c$ nanowires. For a nanowire with an effective thickness and width of 4.3 and 55 nm, respectively, we observe current–voltage characteristics that show direct voltage switching from the superconducting to the resistive state and large current hysteresis over an extended temperature range of up to 18–20 K, as shown in Fig. 1c. One-dimensional phase-slip centers described by the Skocpol–Beasley–Tinkham model[23] can appear in superconducting nanowires with width $W \leq 4.4\xi$[2,24], where $\xi$ denotes the coherence length. In wider nanowires, a phase slippage occurs either by the two-dimensional analog of a phase-slip center (i.e., the order parameter is suppressed across the entire nanowire) or a so-called vortex street (i.e., running phase-slip centers, which are also referred to as kinematic vortices)[25,26]. For our nanowires $W > 40\xi$ and we hence consider the vortex street mechanism as energetically favorable. Note that we will refer to the phase-slip process in our YBCO nanowires using the well-established term "phase-slip line", bearing in mind that the phase slippage occurs due to the motion of kinematic vortices. To observe phase slippage in wide nanowires, as is the case here, it is necessary to realize nanowires with smooth edges and efficient heat removal[26,27]. We calculate the Ginzburg–Landau depairing current density as $J_{GL} = \Phi_0/3^{3/2}\pi\mu_0\lambda^2\xi \approx 390$ MA cm$^{-2}$, where $\Phi_0$ is the magnetic flux quantum, $\mu_0$ is the vacuum permeability, $\lambda = 140$ nm[28], and $\xi = 1.3$ nm[29] are the magnetic field penetration depth and the coherence length at zero temperature, and find that order parameter fluctuations significantly reduce the critical current density to ~20% of $J_{GL}$ for our YBCO nanowires.

To assess the possibility of ELQ in a 55-nm-wide nanowire, we use the extended RCSJ model (Supplementary Note 1). We calculate the zero-bias plasma frequency, $\omega_{p0}/2\pi = (2eI_c/\hbar C_{nw})^{1/2} \approx 1.6$ THz, for such nanowire with capacitance $C_{nw} = 4.8$ fF (Methods section), assuming that the switching current is close to the critical current $I_c$. The energy of corresponding Josephson plasma oscillations $\omega_{p0}\hbar = 6.6$ meV is significantly lower than $2\Delta_0 = 50$–60 meV. Hence, several quantized energy levels can exist in the nanowire. Assuming that the losses are mainly due to thermally equilibrium quasiparticles, we estimate the quality factor of the nanowire in the superconducting state as $Q_s = \omega_{p0}R_{qp}C_{nw} = (2eI_cC_{nw}/\hbar)^{1/2}R_n e^{\Delta/kT} \approx 10^{32}$–$10^{10}$ in the 4–15 K temperature range, where $R_{qp} = R_n e^{\Delta/kT}$ is the quasiparticle resistance, $R_n = 4200$ Ω is the normal-state resistance of the nanowire, $k$ is the Boltzmann constant, and $\Delta = 25$ meV. The exponential temperature dependence of the quasiparticle number in superconductors is only observable until a certain temperature due to the presence of two-level fluctuators and coupling to the environment. Note that in the case of YBCO, the exponential dependence would already provide for an excited state lifetime of 10 years at the temperature of 6.5 K that is sufficient for practical applications.

The registration of the escape from a potential well either by TA or MQT significantly depends on the damping in the resistive

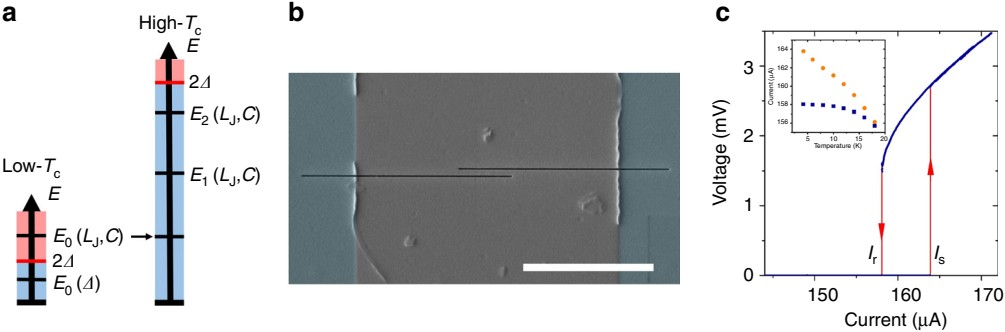

**Fig. 1 YBCO nanowire with phase-slip dynamics. a** Energy diagrams for low-$T_c$ and high-$T_c$ nanowires. Energies below and above $2\Delta$ are shown in blue and red, respectively. **b** Scanning electron micrograph of an YBCO nanowire shaped by two FIB cuts across a microbridge. The YBCO film, STO substrate and FIB cuts are shown in gray, blue, and black, respectively. The scale bar corresponds to 5 μm. **c** IV curve of a 55-nm-wide YBCO nanowire at 4.2 K. Inset: temperature dependences of the average switching $I_s$ (orange circles) and retrapping $I_r$ (blue squares) currents. Source data are provided as a Source Data file.

state. Within the framework of the mechanical analog of the RCSJ model, which considers the motion of a particle in a washboard potential, the system stops in the nearest local minimum after the escape event if the damping is high. The resulting short voltage pulse is very difficult to detect. If the damping is low, the particle continues to move along a tilted washboard potential after a single-escape event, even when the washboard potential has local minima. The phase difference is free running and some finite voltage appears across the JJ, which is straightforward to detect with a voltmeter. The crossover between the low- and high-damping regimes take place at a quality factor of $Q = 0.8382$ (ref. [30]). Here, we use similar criteria to evaluate the behavior of a phase-slip nanowire after an escape event. We calculate the quality factor of the nanowire in the resistive state as $Q_n = (2e/h)(2\Lambda_Q/L)V_s R_{ps} C_{nw} \approx 1$ (Supplementary Note 1), where $R_{ps} = 120$ Ω is the phase-slip line resistance, $V_s = 2.7$ mV is the voltage switching amplitude, and $\Lambda_Q = 200$ nm is the charge imbalance distance[22]. We conclude that the nanowire will switch into the resistive state after a single-escape event because $Q_n > 0.8382$ and the tunneling from different energy levels can be detected by measuring the switching current. Furthermore, we expect a crossover between TA and MQT escape mechanisms at a temperature of $T_{cr} \approx \omega_{p0} \hbar/2\pi k = 12.4$ K[19]. Based on these calculations, we conclude that the energy levels in our YBCO PSNs must be quantized at low temperatures and can be revealed using switching-current measurements.

**Nanowire under equilibrium conditions**. We measured switching-current statistics for a 55-nm-wide 4.3-nm-thick YBCO nanowire, both under equilibrium conditions and under illumination with 77 K black body radiation (BBR) as well as visible light, inducing non-equilibrium states of the wire. In order to reach thermal equilibrium of the nanowire with external radiation, we kept the radiation shield surrounding the nanowire and the nanowire itself at the same temperature. We recorded 1500 IV curves with a current sweep rate of $dI/dt = 0.55$ mA s$^{-1}$ for each of eight distinct temperatures in the 4.2–18 K temperature range, determined the switching-current values and extracted the switching-current distributions (SCDs), which are shown in Fig. 2a. The measured SCDs represent the probability of the decay of the zero-voltage state when the bias current is ramped up. The decay of the zero-voltage state occurs by the TA or MQT with rate that exponentially depends on the bias current[31]. At the highest temperature of 18 K, the SCD shows a single peak at a switching current of 156.12 μA. As the temperature is decreased to 16–14 K, the SCD peak shifts to higher switching currents,

broadens, and develops an asymmetry with a fine structure of closely spaced peaks with spacings of 74 ± 20 nA and 116 ± 31 nA, respectively (enlarged SCDs are shown in Supplementary Fig. 1). Below 14 K, the SCDs show new peaks on either side of the main peak. This three-peaked structure is most pronounced close to 10 K and gradually disappears at 6–4.2 K.

Within the framework of the RCSJ model, the switching and retrapping processes are affected by external noise in a similar way. We measure retrapping-current distributions (RCDs) simultaneously with the SCD and find only a single peak with standard deviation $\sigma_r = 62.8 \pm 3.6$ nA for all temperatures, indicating that the SCD transformation is caused by intrinsic nanowire dynamics rather than by external noise (RCDs are shown in Supplementary Fig. 2).

The only model that predicts the broadening and oscillations of the SCD at temperatures slightly above the TA–MQT crossover temperature $T_{cr} = 12.4$ K was given by Silvestrini et al.[32,33], who considered the dependence of SCDs on the bias current ramp rate. The model predicts that the decay from lower energy levels becomes accessible at temperatures slightly above the crossover temperature, if the bias current ramp rate is high enough, i.e., non-adiabatic. The non-adiabatic transition between the TA and MQT regimes is shown schematically in Fig. 2b–g. At temperatures well below $T_{cr}$, only the lowest (ground) energy level is populated (Fig. 2b). Escape is dominated by MQT from the ground-state energy level and the corresponding SCD is single peaked and narrow (Fig. 2e). When the temperature is close to $T_{cr}$, higher energy levels become populated as a result of thermal fluctuations (Fig. 2c), according to a Boltzmann distribution. If the current sweep rate is high, so that the upper energy level cannot be refilled by thermal fluctuations, then escape from the lower energy levels becomes possible and the nanowire can switch into a resistive state over a wider current range. The corresponding SCD (Fig. 2f) is broad and has a fine structure of closely spaced peaks caused by escape from different energy levels. At temperatures well above the crossover temperature, the energy levels are broadened and their superposition forms a continuous energy band, as shown in Fig. 2d. The nanowire switches into the resistive state via TA from the upper edge of this energy band, resulting in a single-peaked and narrow SCD (Fig. 2g). Non-adiabatic broadening of the SCD can be observed experimentally, when the current sweep rate $dI/dt > I_c/200R_{qp}C$, where $R_{qp}$ is the quasiparticle resistance[32]. By using this expression, we obtain an order-of-magnitude estimate of the quasiparticle resistance, $R_{qp} \approx 3 \times 10^{11}$ Ω and find the quality factor in the superconducting state $Q_s = \omega_{p0} R_{qp} C \approx 1.5 \times 10^{10}$ as well as the lifetime in the excited

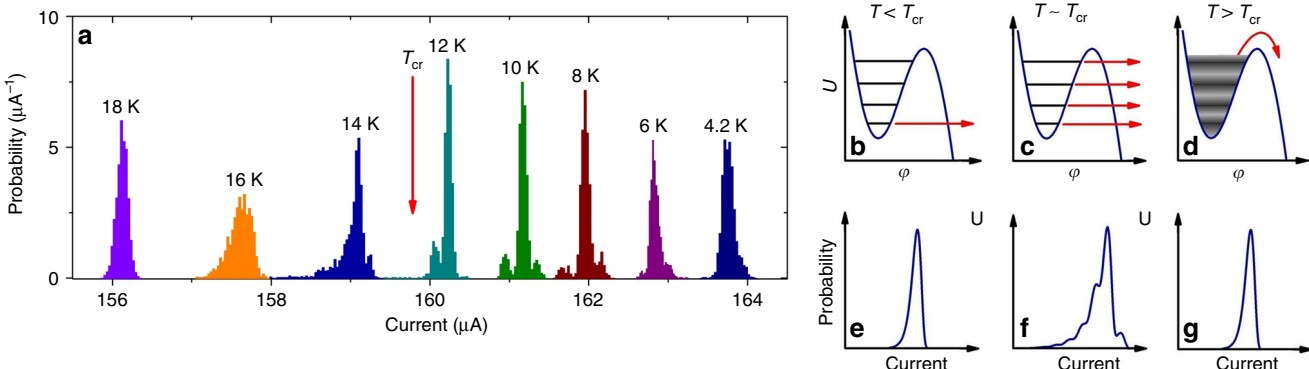

**Fig. 2 Switching-current measurements for the nanowire under equilibrium conditions. a** SCDs for a 55-nm-wide YBCO nanowire. Source data are provided as a Source Data file. **b–g** Washboard potential with quantum states **b–d** and corresponding SCDs **e–f** measured with a non-adiabatic sweep rate at temperatures below, close to, and above the crossover temperature between the MQT and TA regimes.

state $\tau = Q_s/\omega_{p0} \approx 1.5$ ms at $T = 14$–16 K. The quality factor in the superconducting state is in good agreement with our estimates made within the extended RCSJ model. We attribute the presence of smaller peaks on both sides of the main peak in the SCD at $T < T_{cr}$ to interaction of the nanowire plasma oscillations with an external resonant system, i.e., with geometric resonances of the 10-μm-long microbridge or the bow-tie antenna of our device (Methods section), with resonant frequencies of 1.2 THz and 40 GHz, respectively.

In order to probe the energy-level structure in the PSN under equilibrium conditions, we applied external noise to the nanowire via the cables connecting the Dewar insert to the measurement equipment. We found that external noise has a significant effect on the SCDs, as shown in Fig. 3a for three different temperatures at a current sweep rate $dI/dt = 2.75$ mA s$^{-1}$. At 4.2 K, the SCD consists of four nearly equally spaced peaks, which are marked $I_{s1} - I_{s4}$ in Fig. 3a. At 5.9 K, the spacing between the peaks decreases slightly. At 8 K, only two peaks remain in the SCD. The noise-affected RCD shows only a single peak with $\sigma_r = 51 \pm 3$ nA (Supplementary Fig. 3), indicating that the applied low-frequency noise does not cause the multiple peak structure in the SCD, but acts as a trigger for another physical mechanism, which is discussed below.

The nearly evenly paced peaks $I_{s1} - I_{s4}$ in Fig. 3a are signatures of tunneling from different energy levels, as shown in Fig. 3b. If the damping is moderate, i.e., if the quality factor is close to unity, then a particle that escaped from the potential well due to external noise activation (magenta arrow) can be trapped in the upper energy level of the lower potential well. This retrapping process in a PSN is similar to that of thermal or quantum phase diffusion in underdamped tunnel JJs[34,35]. When a particle is trapped in the upper energy level, it can decay to lower energy levels in the same well (blue arrows) or escape from the potential well by tunneling (red arrows), resulting in the presence of multiple peaks in the SCD. Within the framework of this model, the $I_{s1} - I_{s4}$ values are given by a system of equations $\Delta U(I_{si}) = \Delta U_{tun} + E_n(I_{si})$ (1), where i is an integer, and $\Delta U(I_{si})$ and $E_n(I_{si})$ are the energy barrier height and the energy of the populated energy level corresponding to escape at current $I_{si}$, respectively. Here, we assume that the particle tunnels through the barrier when the barrier height for the populated energy level is decreased to $\Delta U_{tun}$. We approximate the nanowire by a harmonic quantum oscillator with energy levels $E_n(I_{si}) = \omega_p(I_{si})\hbar(n + 1/2)$, where $\omega_p$ is the current-dependent plasma frequency. We also use the approximate expressions for barrier height $\Delta U(I) = (hI_c/2\pi e)[(1 - (I/I_c)^2)^{1/2} - (I/I_c)\arccos(I/I_c)]$[36] and plasma frequency $\omega_p(I) = \omega_{p0}[1 - (I/I_c)^2]^{1/4}$ (ref. [19]). By solving the system of

equations (1), we obtain $\omega_{p0}/2\pi = 744$ GHz. We analyze the stability of the solution under small perturbations of the initial parameters and find that the real-valued solution disappears, when the $I_{si}$ values are varied by >100 nA from their initial values. Hence, we conclude that the observed peaks $I_{s1} - I_{s4}$ can indeed be assigned to energy levels in a tilted washboard potential. We note that the calculated zero-bias plasma frequency is only approximately half of our previous estimate based on the nanowire capacitance and the crossover temperature. Using the numerically simulated eigenvalues $E_n/\omega_p\hbar$ of the JJ in a cubic approximation of the washboard potential, which take values of 0.5, 1.45, and 2.37 for $n = 0$, 1 and 2 (ref. [36]), we obtain the $\omega_{p0}/2\pi$ value as 1.5 THz, which is close to our theoretical estimate. The reduced number of peaks in the SCD at 8 K can be attributed to a decrease in energy level lifetime with increasing temperature when transitions to the lower energy level become more probable than tunneling through the barrier.

**Nanowire under illumination with BBR.** In order to probe the energy-level structure of the nanowire using external radiation, we illuminated the device with 77 K BBR, which has a broad continuous spectrum peaked at a frequency of 8 THz and can populate energy levels up to $2\Delta$, resulting in a non-equilibrium state of the nanowire. We studied the resulting non-equilibrium nanowire state in the 5.4–20.1 K temperature range using three current sweep rates of 0.055, 0.55, and 2.75 mA s$^{-1}$. The resulting SCDs (orange bars) are shown in Fig. 4a–d. Well above the crossover temperature, the SCD does not show any sign of quantized energy levels (Fig. 4a), similar to experiments with the nanowire under equilibrium conditions.

When the temperature is close to the crossover temperature (Fig. 4b), the SCD has a single peak at the lowest current sweep rate $dI/dt = 0.055$ mA s$^{-1}$, which broadens toward higher currents with increasing current sweep rate, eventually transforming into a distribution with two peaks at $dI/dt = 2.75$ mA s$^{-1}$. This transformation of the SCD with current sweep rate reflects a transition from adiabatic to non-adiabatic measurements, as lower energy levels become accessible. Below the crossover temperature, at 10.9 K, we observe a strong dependence of the spectral weight of the peak $I_{s2}$ on the current sweep rate, as shown in Fig. 4c. At even lower temperature (5.4 K in Fig. 4d), the spectral weight dependence of the $I_{s2}$ peak with current sweep rate is less pronounced and the $I_{s1}$ peak is hardly observable.

We interpret the difference in spectral weight between the peaks at $I_{s1}$ and $I_{s2}$ in terms of a population inversion resulting from the decay of higher-lying energy levels, similar to

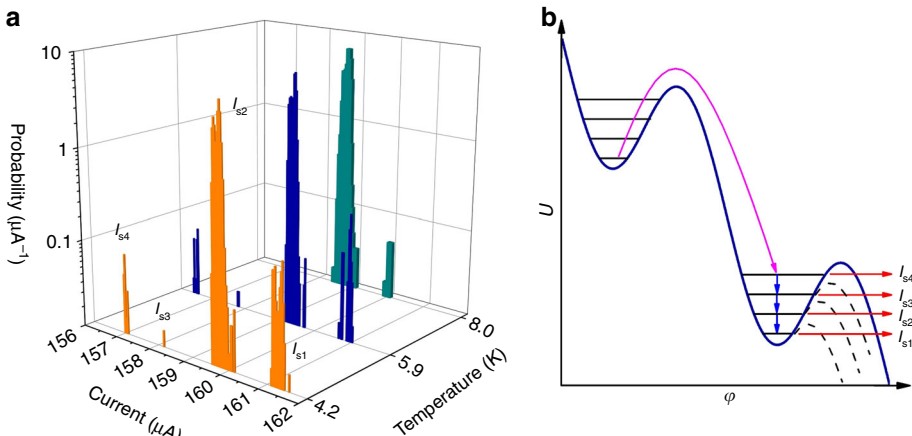

**Fig. 3 Switching-current measurements for the nanowire affected by low-frequency noise. a** SCDs of the 55 nm-wide YBCO nanowire affected by low-frequency noise. Source data are provided as a Source Data file. **b** Noise-induced escape from a local minimum of the tilted washboard potential (magenta arrow) and subsequent escapes into the resistive state by tunneling from the different energy levels (red arrows). Blue arrows correspond to transitions from the upper to the lower energy level. Dashed lines are the washboard potentials at $I = I_{s1} - I_{s3}$.

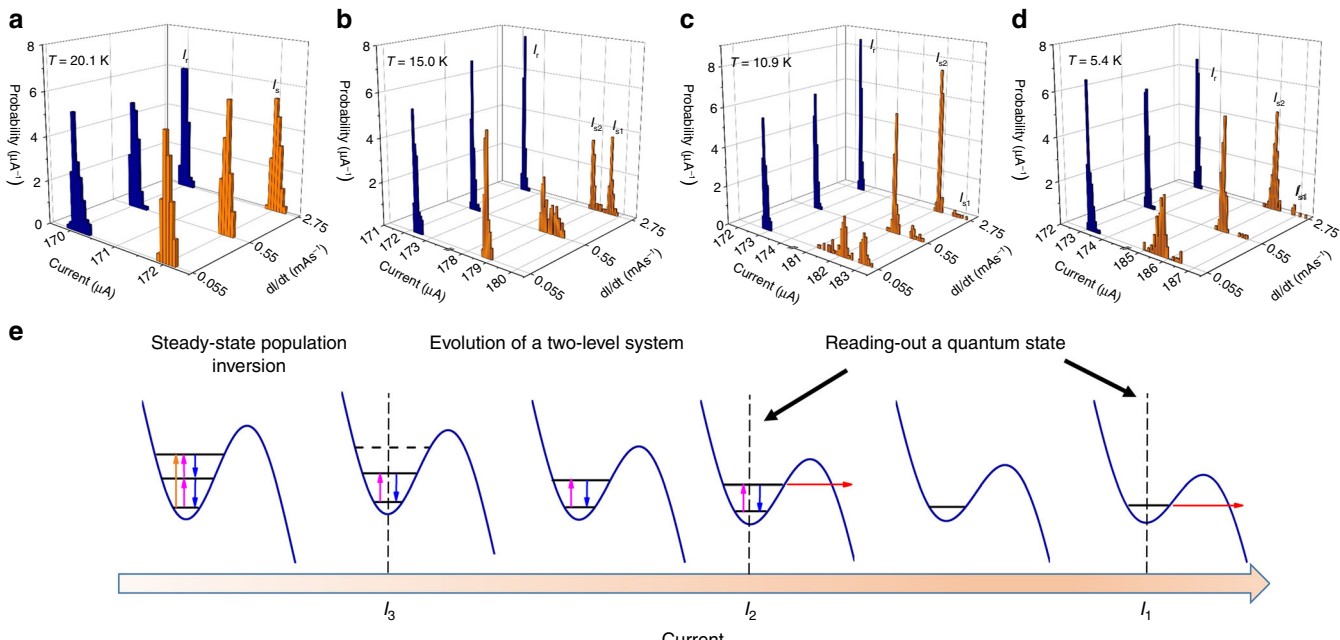

**Fig. 4 Switching-current measurements for the nanowire under illumination with 77 K BBR. a–d** Switching- (orange) and retrapping-current (blue) distributions for a 55-nm-wide YBCO nanowire at different temperatures. Source data are provided as a Source Data file. **e** Energy levels in the local minimum of the tilted washboard potential at different bias currents for the nanowire illuminated using 77 K black body radiation. Transitions between energy levels are shown using orange, magenta, and blue arrows. Escape in the resistive state by tunneling is indicated by red arrows.

observations in superconducting quantum circuits based on tunnel JJs[37,38]. We identify the $I_{s1}$ and $I_{s2}$ peaks in Fig. 4c, d with the ground and first exited energy levels, respectively. The effects of temperature and current sweep rate on the spectral weights of the $I_{s1}$ and $I_{s2}$ peaks can then be assessed using the tilted washboard potential model, as shown in Fig. 4e. Tunneling from the ground, first and second excited energy levels occurs at currents $I_1$, $I_2$, and $I_3$, respectively. Population inversion in the steady state is possible for currents $I < I_3$ when three or more energy levels exist in the potential well. At $I_3 \leq I < I_2$, the number of the energy levels is reduced to two and the nanowire state evolves toward a steady state, with similar population probabilities for the ground and first excited states by spontaneous and

stimulated transitions during the time $t_{32} = [I_3 - I_2]/(dI/dt) \approx [I_2 - I_1]/(dI/dt)$. The nanowire quantum state can be read out at $I = I_2$ or $I_1$. As the energy-level occupation probability depends strongly on temperature and current bias sweep rate, the spectral weight of the SCD peak at $I = I_2$ decreases with increasing temperature or with decreasing current sweep rate.

We calculate the lifetime in the exited state by associating the spectral weight of the $I_{s2}$ peak with the occupation probability of the first excited energy level $P_2(t)$. A fit of the expression $A + B \times \exp(-t/\tau_2)$ to the experimental data (Supplementary Fig. 4) yields reasonable fitting parameters of $A = 0.55$ and $B = 0.39$, and an excited state lifetime of $\tau_2 = 8.3$ ms. Assuming that the lifetime $\tau_2$ at 10.9 K is limited by quasiparticle losses and considering the

nanowire as a lumped element with quasiparticle resistance $R_{qp} = R_n \times e^{\Delta/kT}$, where $R_n \approx 4200$ $\Omega$ is the normal-state nanowire resistance, we obtain a reasonable estimate of the superconducting energy gap in YBCO $\Delta(10.9\,\text{K}) = kT\ln(\tau_2/2\pi CR_n) \approx 17$ meV. The lifetime in the excited state $\tau_2$ at 5.4 K is, however, much longer than $t_{32} \approx 20$ ms, because we only observe a minimal dependence of the population probability $P_2(t)$ of the first exited level on the current sweep rate down to $dI/dt = 55$ $\mu$A s$^{-1}$. The RCDs show only a single peak for all temperatures and current sweep rates, as illustrated by the blue bars in Fig. 4a–d. We find the standard deviations of the retrapping current of 88.1 ± 5.7, 67.9 ± 4.4, and 56.8 ± 5.0 nA at $dI/dt$ of 0.055, 0.55, and 2.75 mA s$^{-1}$, respectively and are able to reproduce similar $\sigma_r$ values with two different experimental setups, confirming that switching into the resistive state originates from internal nanowire dynamics and not external noise.

**Single-photon control of YBCO PSNs.** Additional illumination of the nanowire using optical radiation with an light-emitting diode (LED) led to seemingly counterintuitive results: the number of switching events with higher switching current was observed to increase with increasing LED irradiance. Representative SCDs measured at different irradiances of a blue LED with 460 nm wavelength for a current sweep rate of $dI/dt = 0.55$ mA s$^{-1}$ are shown in Fig. 5a. When the LED is turned off, the nanowire is still subjected to 77 K BBR and switches into the resistive state in current range II (Fig. 5a), which corresponds to tunneling from the first exited state of the nanowire. At an LED irradiance of $I_{LED} = 0.7$ W m$^{-2}$, switching events occur not only in region II but also in region I of the current range in Fig. 5a, which corresponds to tunneling from the ground energy state. As the LED irradiance rises, the number of switching events in current region I increases and some switching events start to appear in the gap between current regions I and II, forming a peak at $I = 186.5$ $\mu$A when $I_{LED} = 8.3$ W m$^{-2}$.

The interaction of optical photons with a superconducting nanowire has been studied widely because of its practical importance for the development of superconducting nanowire single-photon detectors[39–42]. Here, we use a refined hotspot model to analyze single-photon effects in YBCO nanowires[42]. An optical photon whose energy is much higher than the

superconducting energy gap disrupts tens of Cooper pairs, resulting in the appearance of non-equilibrium quasiparticles. The absorbed photon induces a normal-state domain (hotspot) across the nanowire when the number of non-equilibrium quasiparticles reaches $N_q = n_s W d(\pi D \tau_{th})^{1/2}(1 - I/I_c)$, where $n_s$ is the local density of paired electrons, $d$ is the nanowire thickness, $D$ is the quasiparticle diffusion coefficient, and $\tau_{th}$ is the quasiparticle thermalization time[42]. We consider the quasiparticle motion during the thermalization process to be diffusive because the electron–electron scattering time $\tau_{e-e} = 0.1$ ps[43,44] in YBCO is much shorter than the thermalization time $\tau_{th} = 0.56$ ps[43,45]. If the entire photon energy $E_{ph}$ is transferred to the quasiparticles, their actual number is given by $N_q = E_{ph}/\Delta$. The boundary for hotspot appearance can then be calculated as $I_{HS} = I_c - [j_c E_{ph}/n_s \Delta (\pi D \tau_{th})^{1/2}]$, where $j_c$ is the critical current density. Photon absorption below and above $I_{HS}$ has qualitatively different consequences. For $I < I_{HS}$, the photon creates non-equilibrium quasiparticles, but the normal-state domain across the nanowire does not appear. For $I > I_{HS}$, photon absorption results in a hotspot across the nanowire, which leads to local collapse of the order parameter. The PSN evolves from this transient state toward a state with a phase-slip process, corresponding to switching of the nanowire from the superconducting to the resistive state, as described in ref. [46].

For photons of wavelength 460 nm, we calculate $I_c - I_{HS} = j_c E_{ph}/n_s \Delta (\pi D \tau_{th})^{1/2} = 1.1$–1.3 $\mu$A using $n_s = 1.1 \cdot 10^{27}$ m$^{-3}$ (ref. [47]), $\Delta = 25$–30 meV, $D = 20$ cm$^2$ s$^{-1}$ (ref. [44]), $\tau_{th} = 0.56$ ps[43,45], $E_{ph} = 2.7$ eV, and the experimentally measured $j_c = 78.6$ MA cm$^{-2}$. In Fig. 5a, we highlight this current region III in red, in which the nanowire can switch to the resistive state only by the 460 nm wavelength photons. We assume that the critical current in the refined hotspot model corresponds to the switching current of the nanowire in the ground energy state.

Figure 5b illustrates schematically the interaction of optical photons with a PSN that has quantized energy levels and is prepared in the excited state. When the LED is turned off, the nanowire switches to a resistive state at current $I_2$ due to tunneling from the first excited energy state. At a low LED irradiance ($I_{LED} = 0.7$ W m$^{-2}$ in Fig. 5a), the nanowire can absorb the photon before reaching current $I_2$. Since photon absorption takes place at $I < I_{HS}$, it creates a number of

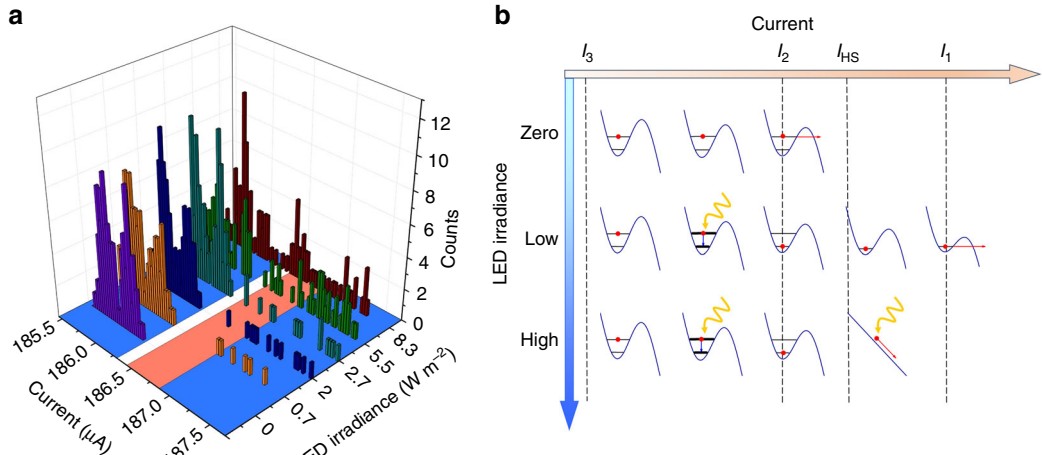

**Fig. 5 Switching-current measurements for the nanowire illuminated by optical radiation with 460 nm wavelength. a** SCDs of a 55-nm-wide YBCO nanowire measured at 5.4 K for different LED irradiances. Regions I and II (blue) correspond to ground and exited energy states of the nanowire. Region III (red) is the current region, in which switching can only occur by the hotspot effect. Source data are provided as a Source Data file. **b** Illustration of the interaction of optical photons with the YBCO nanowire. The populated energy level is shown by a red dot. Photon absorption and escape events are indicated by wavy yellow and red arrows, respectively. Collapse of the order parameter after photon absorption is shown by an inclined blue line. The currents $I_1$, $I_2$, and $I_3$ are the same as in Fig. 4e. The current $I_{HS}$ indicates the boundary of the hotspot regime.

non-equilibrium quasiparticles, resulting in an increase in nanowire losses, which is shown in Fig. 5b by a broadening of the energy levels and, hence, faster decay from the excited to the ground energy state. Switching into the resistive state then occurs at the higher current $I_1$ by tunneling from the ground energy level. Therefore, the nanowire quantum state can be changed by single-photon absorption.

When the radiation irradiance is increased (corresponding to high LED irradiance in Fig. 5b, and $I_{LED} = 1.4–5.5$ W m$^{-2}$ in Fig. 5a), the nanowire can absorb a second photon at bias currents of $I > I_{HS}$, where hotspot conditions are met. As a result of the local collapse of the order parameter, the oscillating phase-slip process appears and the nanowire switches into the resistive state. Since the absorbed photon changes the number of phase-slip processes in the nanowire, i.e., the phase-slip state of the nanowire, we refer to this process as the phase-slip mechanism for photon detection.

At very high radiation irradiances, switching of the nanowire occurs predominantly at the boundary of the hotspot region, forming a peak in the SCD at $I = I_{HS}$. We observe this very high irradiance regime at $I_{LED} = 8.3$ W m$^{-2}$ in Fig. 5a, where a peak at the edge of region III is visible. We treat the observed interaction of the optical photons with the YBCO nanowire as a single-photon process because the position of the photon-induced SCD peak corresponds to the single-photon energy and the radiation power is much lower than that required for the two-photon process. Comparing the LED irradiance to the $W = 55$ nm wide and $L = 2$-μm-long nanowire, we estimate that no more than $N = (\eta L W I_{LED})/E_{ph} \approx 53{,}000–630{,}000$ photons s$^{-1}$ are absorbed by the nanowire, where $I_{LED} = 0.7–8.3$ W m$^{-2}$ is the LED irradiance, and $\eta \approx 0.3$ is the theoretical estimate for the absorption coefficient of a metal film, which we calculate from the YBCO film sheet resistance $R_s = 115$ Ω at temperatures just above $T_c$[48]. The average photon absorption rate hence is in the range of one photon per 1.6–19 μs. This time interval is approximately six orders of magnitude longer than the quasi-particle recombination time, $\tau_r = 3–10$ ps, in optimally doped YBCO thin films on STO substrates or single-crystal samples[49,50]. The perturbation caused by the absorption of a photon will thus decay long before a subsequent photon will be absorbed in the nanowire. Based on the above, we consider our nanowire to be in the single-photon regime where multiphoton processes occur with extremely low probability and can be neglected.

In this work, we have implicitly considered a nanowire made from YBCO, which has d$_{x2-y2}$-wave symmetry of the order parameter as a fully gapped superconductor. We find that a small ($N_q \leq 100$) number of non-equilibrium quasiparticles generated by an optical photon is significantly larger than the number of equilibrium quasiparticles in the nanowire, resulting in fast decay from the excited to the ground state. This behavior is expected for a fully gapped superconductor with an exponentially small number of equilibrium quasiparticles at a temperature well below the critical temperature. Deviations from d$_{x2-y2}$-wave symmetry in our YBCO nanowires can arise from size or doping effects, which have recently been observed in different cuprate superconductors[51,52].

## Discussion

In summary, we have fabricated sub-100-nm-wide YBCO nanowires with phase-slip dynamics and measured their switching-current statistics under equilibrium and non-equilibrium conditions. Our experimental data show ELQ in YBCO PSNs. The YBCO nanowires have a high crossover temperature between TA and quantum regimes of 12–13 K and their lifetime in the excited state exceeds 20 ms at 5.4 K, which is at least one order-of-magnitude longer than in

low-$T_c$ tunnel JJs[1]. We also show that the absorption of a single-photon changes the quantum and phase-slip states of YBCO nanowires. Our findings demonstrate that phase-slip YBCO nanowires are promising systems for quantum technology applications, including quantum sensing and computing.

## Methods

**Nanowire fabrication**. YBCO nanowires were fabricated from an 8.2 nm (7 unit cell) thick YBCO film deposited on a TiO$_2$-terminated (100) STO substrate by dc sputtering at high (3.4 mbar) oxygen pressure. YBCO deposition followed a procedure that is described elsewhere[53]. A total of 100-nm-thick Au contact pads were deposited ex situ using room temperature dc magnetron sputtering with a shadow mask. Following contact pad deposition, nanowires were fabricated in a two-stage process. In the first stage, 10-μm-wide 10-μm-long microbridges integrated with a bow-tie antenna and leads were patterned using optical UV contact lithography with a PMMA resist and Ar ion beam etching. In the second stage, 2-μm-long nanowires aligned along STO crystallographic axes were fabricated across the microbridges with two cuts made with FIB milling using a Au/PMMA protective layer. A sketch of the device is shown in the Supplementary Fig. 5a. More details on the patterning process can be found in ref. [22].

**Experimental setups**. We performed the measurements using two experimental setups. The first experimental setup was based on a liquid helium storage Dewar insert filled with He exchange gas, in which the nanowire and surrounding radiation shield had the same temperature. The second setup was based on an HLD-5 liquid helium cryostat (Infrared Laboratories, Inc.). The sample was placed on a sample holder mounted on the 4 K stage of the cryostat and shielded by a radiation shield with a quartz window. The sample was illuminated through the window using continuous optical radiation emitted by LEDs and 77 K BBR from a 77 K radiation shield. The LEDs were placed in front of the window at a distance of 2 cm from the substrate and cooled to 77 K. The sample holder temperature was maintained to an accuracy of ±5 mK at 4.2 K and ±20 mK at 20 K for the Dewar-insert-based setup, and ±10 mK for the cryostat-based setup over the whole temperature range. The wiring inside the Dewar insert was made with twisted-pair cables and had a bandwidth of 3.9 MHz. The nanowires inside the cryostat were connected to room temperature measuring equipment using high-frequency SMA coax cables and a 10 GHz probe. We used battery-operated low-noise analog electronics with a 100 kHz frequency bandwidth to sweep the bias current and amplify the voltage across the nanowire. The root-mean-square noise of the current source was 1 nA. The output signals of the analog electronics, proportional to the current and voltage across the nanowire, were digitized using a simultaneous 16-bit data acquisition board DT9832 (Data Translation). All electrical connections, apart from the HF coax cables, were filtered using low-frequency feedthrough filters. Cables between the analog electronics and cryogenic units were as short as possible to eliminate electromagnetic interference. The frequency spectrum of the output voltage signal was controlled before the measurement to ensure that no low-frequency external noise was present in the measurement system.

**Switching-current measurement**. In order to measure the switching-current and retrapping-current statistics, the bias current through the nanowire was ramped linearly up and down over the range (0.85–1.05) $I_s$ and $IV$ curves were recorded with $2 \times 10^4$ points per curve. The spacing between independent current points of 15.3 nA was determined by the 16-bit resolution of the data acquisition board. The switching and retrapping currents were determined by post-processing of the recorded data. Standard deviations of the retrapping current were computed in the standard way.

**Nanowire capacitance calculation**. The nanowire layout shown schematically in the Supplementary Fig. 5b is similar to that of a coplanar waveguide with ground. As a result of this similarity, we used a coplanar waveguide calculator[54] to calculate the nanowire capacitance. The following parameters were used for the capacitance calculations: spacing $S = 200$ nm, substrate thickness $H = 1$ mm, and dielectric index of the STO substrate $\varepsilon = 300$. The spacing $S$ consisted of the 60-nm-wide FIB cut and two 70-nm-wide damaged insulating regions of YBCO on the sides of the FIB cut[22]. By using the coplanar waveguide calculator[54], we obtained a nanowire specific capacitance per unit length of $C_S = 2.4$ fF μm$^{-1}$ and a total capacitance of the 2-μm-long nanowire of $C = 4.8$ fF.

## Data availability

The source data underlying Figs. 1c, 2a, 3a, 4a–d, 5a and Supplementary Figs. 1–4 are provided as a Source Data file. Data supporting the findings of this manuscript are available from the corresponding author upon reasonable request.

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

## Acknowledgements

M.L. thanks G. Catelani for valuable discussions. This work was partially supported by ER-C Project No. C-088. C.S. acknowledges financial support from the Ministry for Innovation, Science and Research (North Rhine-Westphalia).We thank the Münster Nanofabrication Facility (MNF) for technical support during device fabrication.

## Author contributions

M.L. and M.K. fabricated the nanowires. M.L. and M.A.W. performed the measurements. S.F. and I.G. contributed to the experiments. M.L., I.G., R.E.D.B., and C.S. co-wrote the paper. All authors commented on the manuscript.

## Competing interests

The authors declare no competing interests.
