## [Peer Review File · Nature Communications]

Reviewers' comments:

Reviewer #1 (Remarks to the Author):

The manuscript presents an interesting set of very novel experiments on high- T_c superconducting nanowires. The results indicate that high- T_c superconducting nanowires can act quantum device and their energy states are quantized. The quantization is seen at a relatively high temperatures of several Kelvins. This is in stark contrast with the usual experiments on Al qubits, in which macroscopic quantum behavior emerges only at millikelvin temperatures. Thus, this work might lead to a development of a new area of quantum devices operating above the temperature of liquid helium. The results are important and should be published.

The following remarks need be addressed before the publication:

1. There are two types of estimates of the quality factor. They differ by 10 orders. Explain the physical meaning of this fact. The capacitor values used for these two different estimates were different, I think. Why the capacitance is taken differently.
2. Explain the role and the significance of the capacitance between the micron-scale superconducting electrodes connected to the nanowire.
3. Provide a possible physical picture of the excited states in the nanowire. What are these excitations? Quantized plasma waves perhaps?
4. It is not clear if all the results are obtained on a single wire or on many 55 nm wide wires. How many samples have been tested?
5. How many times the V-I curve was measured in order to produce a typical distribution?
6. Line 220: "...are able to reproduce similar ... values with two different experimental setups, confirming that switching into the resistive state originates from internal nanowire dynamics and not external noise." If two measurements have been done on different setups, then please show both results for comparison, at least in the supplementary materials.

Reviewer #2 (Remarks to the Author):

The authors claim the observation of quantum phase slip and energy quantization in YBCO nanowire. The sample has a finite critical current with hysteresis, and $I_c R$ product of about 3mV.

The data probably reflect some macroscopic quantum tunneling with quantized energy level physics. However, the analysis of the result did not follow the physics of quantum phase slip, which is the main claim of the manuscript. Instead, the entire analysis was done based on Josephson effect, with the washboard potential in phase space.

If their analysis is correct in some way, then very likely the nanowire contains a Josephson junction that dominates the transport. In this case, they observed MQT of YBCO Josephson junction, not phase slip nanowire. If they assume a JJ with 120 Ohm (from fig1c, it looks more like 20 Ohm) normal resistance and several fF capacitance, which is typical for JJs of their size, they will have plasma frequency of what they observed. On the other hand, MQT (not quantized energy level) in high T_c oxide material in Josephson regime was first demonstrated by Inomata et al PRL 95, 107005 (2005), but they fail to mention it.

Concerning the claimed 1.5 ms lifetime of the excited states, I am not so sure where did they get the value $R_{qp} > 1.5 \times 10^{10}$ Ohm in order to obtain the lifetime. From figure 1c, the resistance is near 20 Ohm. The experimentally obtained line width in fig 2a gives a much shorter time constant.

Based on these observations, and mainly because of the inconsistency of the claim of quantum phase slip and the Josephson junction model used for the analysis, I don't think it should be published in Nature Communications.

Reviewer #3 (Remarks to the Author):

The main advance on previous light/induced phase slip experiments appears to be that single photons are used to change state, and the resulting resistance change is measured statistically. Without doubt the paper is potentially interesting to readers of Nature communications, particularly the realization of single photon detection using SC nanowires. However, I am not convinced about the applicability of the RCSJ model to the present experiments - or at least the way it is presented.

I am concerned about the interpretation (e.g. Fig. 2) in terms of quantised levels. How is it justified to consider that a 50 nm wide and 5 nm thick YBCO strip is a quasi-1D nanowire? The in-plane coherence length in this material is only 2 nm in plane, and <1 nm out of plane? Are there multiple channels? How do they function?

If this is a SC nanowire, then one should observe steps in the CVCs corresponding to different configurations of PSC along the wire. Why are they not observed? Are the sidebands due to this perhaps?

The temperature dependence also doesn't make sense to me: within a quantised level picture, if the current is ramped, then above a threshold value, switching to a resistive state would be expected at the ground state value (which is always the lowest, at any temperature). Why is the ground state shifting with T? To observe a T-dependent shift of the current, the shunt resistance would need to be temperature dependent, which would change the potential also.

Another issue is the discussion of the photon excitation process. The discussion of the subsequent quasiparticle dynamics is based on the assumption that QP diffusion is dominant. This is not true in high-T_c superconductors, where phonon escape determines the QP lifetime, according to the Rothwarf-Taylor mechanism, particularly in thin films. This has been discussed in the literature quite extensively for YBCO.

In summary, the paper as it stands cannot be recommended. However, I don't think it's beyond repair.

Other comments and suggestions:

The title might want to mention single photon control of phase slips.

In the abstract it is emphasised that the improvement for using YBCO nanowires seemingly have a longer lifetime and remain in the quantum regime at higher temperature than low-T_c nanowires. This is not particularly surprising, considering the larger gap and resulting change in QP recombination dynamics, and distracts the reader from the main issues.

Other comments and suggestions:

In the abstract it is emphasised that the improvement for using YBCO nanowires seemingly have a longer lifetime and remain in the quantum regime at higher temperature than low-T_c nanowires. This is not particularly surprising, considering the larger gap and resulting change in QP recombination

dynamics, and distracts the reader from the main issues.

The title might want to mention single photon control of phase slips, which is after all the most interesting aspect of the data.

Response to reviewer 1

The authors would like to thank the reviewer for his/her valuable comments, which helped to improve the manuscript. Our response follows below (the reviewer's comments are given in italic)

The manuscript presents an interesting set of very novel experiments on high- T_c superconducting nanowires. The results indicate that high- T_c superconducting nanowires can act quantum device and their energy states are quantized. The quantization is seen at a relatively high temperatures of several Kelvins. This is in stark contrast with the usual experiments on Al qubits, in which macroscopic quantum behavior emerges only at millikelvin temperatures. Thus, this work might lead to a development of a new area of quantum devices operating above the temperature of liquid helium. The results are important and should be published.

We appreciate the positive feedback from the reviewer. The revisions made according to the reviewer's comments are listed below and in the summary of changes and highlighted in the manuscript.

The following remarks need be addressed before the publication:

- I. There are two types of estimates of the quality factor. They differ by 10 orders. Explain the physical meaning of this fact. The capacitor values used for these two different estimates were different, I think. Why the capacitance is taken differently.*

Reply: We have included a more detailed description of the extended RCSJ model in the "Supplementary Information" that should help to explain the electrostatics of the YBCO phase-slip nanowire in greater clarity. Within the extended RCSJ model we estimate the quality factors in the superconducting and the resistive states: these differ by 10 orders of the magnitude because of the large difference in the nanowire resistance in the normal and superconducting states. Note that, our model is very similar to the RSJN model ¹ developed for the unshunted tunnel Josephson junctions where the "leakage" resistance in the superconducting state is significantly larger than the normal-state resistance.

We here consider the nanowire shaped by two FIB cuts as a coplanar waveguide where the capacitance is distributed along the nanowire, which acts as a central electrode. We assume that in the resistive state the high-frequency currents flow mainly in the resistive part of the nanowire due to the impedance mismatch between the resistive and superconducting parts. Therefore, in the resistive state only the part of the distributed nanowire capacitance, which corresponds to the resistive part of the nanowire, is charged during the phase slip. The explanation of the difference in the capacitance values in superconducting and resistive states is now included in the section “Electrodynamics of YBCO phase-slip nanowires” (Supplementary information).

II. Explain the role and the significance of the capacitance between the micron-scale superconducting electrodes connected to the nanowire.

Reply: The capacitance is required to induce the Josephson plasma oscillations. In the superconducting state, the capacitance controls the energy of the Josephson plasma oscillations, which makes the existence of the quantized energy levels possible and helps to establish a “running” state in the resistive state.

The role and the significance of the capacitance are now explained in a more detailed description of the extended RCSJ model (Supplementary Information) that we used to describe the electrodynamic of the YBCO phase-slip nanowire.

III. Provide a possible physical picture of the excited states in the nanowire. What are these excitations? Quantized plasma waves perhaps?

Reply: In our manuscript we assume that the excited states in the YBCO nanowire are quantized Josephson plasma oscillations. However, a coexistence of other plasma modes such as Carlson-Goldman ² or Mooij-Schön ³ plasma modes could not be excluded based on our current experimental data.

In order to avoid speculative statements, we refrain from discussing the physical picture of the excited states in the nanowire. We plan to investigate these states in greater detail in a forthcoming publication featuring more sophisticated experiments, which were beyond the scope of the current manuscript.

IV. *It is not clear if all the results are obtained on a single wire or on many 55 nm wide wires. How many samples have been tested?*

Reply: We fabricate many chips with eight nanowires per each chip, however we systematically varying design parameters (such as the nanowire width) from one device to another. For the switching-current measurements we chose the nanowires with effective widths of 55, 85, 95 and 160 nm that have the largest current hysteresis. All sub100-nm-wide nanowires demonstrated switching current distributions with discrete peaks, as shown in figure 4. An illumination of these nanowires with optical radiation resulted in an increase of the high-current peaks in the SCD, as shown in figure 5. The noise-induced SCDs presented in figure 3 were obtained for the 55-nm-wide nanowire. Among these four nanowires, a 55-nm-wide nanowire had the largest temperature range where the current hysteresis was observed that made possible to clearly observe the transition from TA to MQT regime.

V. *How many times the V-I curve was measured in order to produce a typical distribution?*

Reply: During experiments with a dip stick, we recorded 1500 IV curves (line 137 in the manuscript) to calculate SCDs and RCDs. During experiments with the HLD-5 cryostat, we recorded a few hundred IV curves to calculate SCDs and RCDs. The smaller statistics of the data collected with the HLD-5 cryostat is due to longer acquisition times (up to 2 sec per IV curve) and shorter liquid helium holding times. We used short HF coax cables to avoid interference, which however results in a high heat load at the 4K stage.

VI. *Line 220: "...are able to reproduce similar ... values with two different experimental setups, confirming that switching into the resistive state originates from internal nanowire dynamics and not external noise." If two measurements have been done on different setups, then please show both results for comparison, at least in the supplementary materials.*

Reply: In our experiments, we measure RCDs to be sure that the SCDs are not influenced by the external noise. If external noise dominated, the RCD would necessarily be of (symmetric) Gaussian shape and dependent on the specifics of the experimental setup. This was not observed. The retrapping-current distributions obtained with the dip stick and HLD-5 cryostat are presented in Figure S2, Figure S3 and Figure 4. Enlarged RCDs obtained with HLD-5 cryostat can be included in the Supplementary Information if desired.

The revisions made in response to the reviewer's comments are in the summary of changes and highlighted in the manuscript.

References

- 1 Likharev, K. K. *Dynamics of Josephson Junctions and Circuits*. (Gordon and Breach, 1986).
- 2 Carlson, R. V. & Goldman, A. M. Propagating Order Parameter Collective Modes in Superconducting Films. *B Am Phys Soc* **20**, 58-58, (1975).
- 3 Mooij, J. E. & Schon, G. Propagating Plasma Mode in Thin Superconducting Filaments. *Physical Review Letters* **55**, 114-117, (1985).

Response to reviewer 2

The authors would like to thank the reviewer for his/her valuable comments, which helped to improve the manuscript. Our response follows below (the reviewer's comments are given in italics).

The authors claim the observation of quantum phase slip and energy quantization in YBCO nanowire. The sample has a finite critical current with hysteresis, and $I_c R$ product of about 3mV.

Reply: We would like to emphasize that the sample reported in the manuscript is a nanowire rather than a Josephson junction! Therefore, the voltage switching amplitude V_s , which the reviewer consider by analogy with a Josephson junction as an $I_c R_n$ -product, has very different meaning here. The voltage switching amplitude is determined by the nanowire width and the velocity of the kinematic vortices^{1,2} rather than the Josephson dynamics.

1. The data probably reflect some macroscopic quantum tunneling with quantized energy level physics. However, the analysis of the result did not follow the physics of quantum phase slip, which is the main claim of the manuscript. Instead, the entire analysis was done based on Josephson effect, with the washboard potential in phase space.

Reply: We acknowledge that the use of the RCSJ model for the analysis of the electrostatics of a phase-slip YBCO nanowire may not be obvious at first glance. To clarify the appropriateness of our analysis we here offer an extended explanation, which is also reflected in condensed form in the revised manuscript. Importantly, we would like to clarify that our analysis is based on the formalism of the RCSJ model, not on the Josephson effect in 1D weak links. Below we show that the electrostatics of a YBCO phase-slip nanowire can be described by the RCSJ model with minor modifications that account for the differences between phase-slip nanowires and Josephson junctions.

Indeed, the RSJ and later RCSJ models were developed to describe the electrostatics of 1D superconducting weak links where the Josephson effect occurs. However, the main approximations of the RCSJ model are of very general nature. Within the framework of the

RCSJ model³⁻⁵, the total current through the Josephson junction is considered as the sum of a supercurrent I_s , a normal current I_n , a displacement current I_d , and a current fluctuations $\delta I(t)$ as

$$I + \delta I(t) = I_s + I_n + I_d \quad (1)$$

Further, it is assumed that the junction has resistance R and capacitance C , which are voltage-, frequency-, and temperature-independent, such that $I_n = V/R$ and $I_d = C \cdot dV/dt$. The dependence of the superconducting current on the phase difference ϕ for the Josephson junctions is given by $I_s = I_c \sin(\phi)$, where I_c is the fluctuation-free critical current. Using the voltage-phase Josephson relationship $\hbar(d\phi/dt) = 2eV$, equation (1) can be rewritten as

$$I + \delta I(t) = I_c \sin(\phi) + (\hbar/2e)(1/R)(d\phi/dt) + (\hbar/2e)C(d^2\phi/dt^2) \quad (2)$$

where \hbar is Planck's constant and e is the electron charge.

For structures with a tunnel conductivity, equation (1) is a valid approximation for voltages $V < 2\Delta/e$ and for frequencies $f < 2\Delta/h$ ⁶. In the case of nanowires with direct conductivity, equation (1) can be used to approximate the “two-fluid” model, which appropriately describes our YBCO nanowires as long as the frequencies of all characteristic processes are small compared to $2\Delta/h$ ⁷ and there are no vortices in the nanowire.

First, we consider the superconducting state of the YBCO nanowire with phase slips caused by fluctuations of the order parameter and show that it can be described by equation (2).

The voltage-phase Josephson relationship $\hbar(d\phi/dt) = 2eV$ is valid for our YBCO nanowires, at least in the superconducting state³.

Based on the geometry of our 2- μm -long nanowire on an STO substrate we expect a resonance at a frequency of 6.1 THz, which is close to $\Delta/e = 6$ THz, where Δ is the superconducting energy gap. At Josephson plasma frequencies $\omega_p/2\pi \ll \Delta/e$, which are of practical interest in here, the nanowire can be considered as a lumped element where the normal and displacement currents can be expressed as $I_n = V/R_{nw} = (\hbar/2e)(1/R_{nw})(d\phi/dt)$ and $I_d = C_{nw}dV/dt = (\hbar/2e)C_{nw}(d^2\phi/dt^2)$, respectively. Here C_{nw} is the nanowire capacitance and R_{nw} is the nanowire resistance. In the superconducting state R_{nw} is equal to the quasiparticle resistance R_{qp} of the nanowire. This replacement is similar to that in the RSJN model of the unshunted Josephson junction where the junction resistance is set equal to the “leakage” resistance at $V < \Delta/e$ vs. the normal-state resistance at voltages $V > \Delta/e$ ⁶. Hence, the expressions for the normal and displacement currents resemble those in equation (2).

The current-phase relationship (CPR) of a “conventional” superconducting nanowire has a characteristic “sawtooth” shape in striking difference from the sinusoidal CPR employed in the RCSJ model ⁷. However, if fluctuations of the order parameter cannot be neglected, the nanowire CPR will differ from such “sawtooth” shape ^{8,9}. Within the framework of the oversimplified model of a phase-slip nanowire, we can assume that at any given moment in time the critical current of the nanowire is determined by a segment of the nanowire with length $L \approx \xi$ in which the order parameter is reduced because of fluctuations, while in the rest of the nanowire the order parameter can be considered as undisturbed. The critical current density of the phase-slip nanowires measured for our YBCO samples is approx. 20% of the depairing current, such that the order parameter in the regions neighboring the segment with reduced order parameter is not significantly disturbed by the transport current. Note that the characteristic time of the order parameter fluctuation, i.e. $2\Delta/h$, is much shorter than the Josephson plasma frequency. Hence, the YBCO nanowire can be considered as a short S-S'-S weak link that stochastically appears in the nanowire at any given moment in time, for which a sine-like CPR is expected ¹⁰.

Matveev et al. ⁸ and Khlebnikov ⁹ have studied the CPR in superconducting nanowires with strong fluctuations in theoretical works. Matveev et al. predicted that in those nanowires where the fluctuation of the superconducting order parameter cannot be neglected the CPR transforms from a “sawtooth” to sinusoidal shape ⁸. This transition has experimentally been observed in Ref. [11]. Recently, it was shown by Khlebnikov that the activation barrier in the phase-slip nanowires is equal to $\varepsilon = 2^{1/2}(1-i_b)^{1/2} + O[(1-i_b)^{3/2}]$, where $i_b = I/I_c$, and hence, at $i_b \rightarrow 1$ the activation barrier scales as $(1-i_b)^{3/2}$ ⁹. This $(1-i_b)^{3/2}$ -scaling of the activation barrier is characteristic for Josephson junctions with sinusoidal CPR. Hence, even for the phase-slip nanowires, where the complete transformation of the CPR from “sawtooth” to sinusoidal does not generally occur, one can expect the sinusoidal CPR at currents close to the critical current. In this regard Arpaia et al. showed that the CPR of a 50-nm-wide and 50-nm-thick YBCO nanowire is sine-like ¹², meaning that it undergoes at least a partial transformation from “sawtooth” to sinusoidal shape. Therefore, we assume that the CPR of our 55-nm-wide and only 5-nm-thick nanowire, i.e. 10 times smaller cross section than the nanowire in Ref. ¹², is also of sinusoidal shape, at least at currents close to the critical current $I_s(I \rightarrow I_c) = I_c \sin(\varphi)$, where our experiments were performed.

From the above discussion, we conclude that the electrodynamics of the YBCO phase-slip nanowire in the superconducting state can be described by an equation similar to that of equation (2) of the RCSJ model. Hence, we apply the formalism of the RCSJ model for calculating the plasma frequency and the quality factor of phase-slip YBCO nanowires in the superconducting state.

On the other hand, the resistive state of a nanowire that is shunted by a large capacitor resembles an underdamped Josephson junction. In this case, the escape from one local potential minimum to another lower-lying potential minimum changes the phase across the junction by 2π and charges the capacitor, which then stimulates further phase evolution in accordance with the voltage-phase Josephson relationship $d\phi/dt = (2e/h)V$. Thus, a single escape event switches the underdamped junction into the resistive “running” state. Once the phase of the nanowire shunted by a large capacitance slips by 2π after the escape from a potential well, the phase change leads to charging of the capacitor, which stimulates a subsequent phase slip by the stored charge. Hence, a single escape event can switch the capacitor-shunted nanowire into a resistive state similar to an underdamped Josephson junction.

However, there are of course important distinctions between “running” states in Josephson junctions and phase-slip nanowires. While in a Josephson junction the discharge rate of the capacitor is controlled by the normal-state resistance of the Josephson junction, R_n , which determines the crossover between the overdamped and underdamped regimes at a quality factor of $Q = \omega_p R_n C \approx 0.8382$ ¹³, the discharge rate of the capacitance of a nanowire is controlled by the resistance of the phase-slip center (or line), which can be determined from a corresponding IV-curve as $R_{ps} = dV/dI$ ⁷. A second distinction concerns resistive states of the Josephson junction and phase-slip nanowire. At the currents above the critical current, only a part of the nanowire with length of $2\Lambda_Q$, where Λ_Q is the charge imbalance distance, switches in the resistive state. Hence, the current induced by high-frequency phase-slip oscillations flows only in the resistive part of the nanowire with length of $2\Lambda_Q$, because of the impedance mismatch between the resistive and superconducting part of the nanowire at the frequency of the phase-slip oscillations, and only a part of the nanowire capacitance with a value of $(2\Lambda_Q/L)C_{nw}$ will actually be charged. Then, the corresponding quality factor of the nanowire in the resistive state is equal to $Q_n = (2\Lambda_Q/L)\omega_{ps}R_{ps}C_{nw}$, where $\omega_{ps} = (2e/h)V_s$ is the frequency of the phase-slip oscillations. A third distinction between “running” states in Josephson junctions and phase slip nanowires is that the

capacitance of wide ($W \gg \xi$) nanowires cannot generally be estimated from the I_f/I_s ratio, because here also the retrapping current is limited by the time constant of the phase-slip oscillations τ_{ps} , which have to be shorter than the phase relaxation time¹⁴.

We summarize the above differences between the superconducting and resistive state in the following table:

	Superconducting state	Resistive state
Characteristic frequency	Josephson plasma frequency $\omega_p = (2eI_c/\hbar C_{nw})^{1/2}$	Frequency of phase-slip oscillations $\omega_{ps} = (2e/h)V_s$
Losses	Quasiparticle resistance $R_{qp} = R_n e^{\Delta/kT}$	Resistance of the phase-slip line $R_{ps} = dV/dI$
Quality factor	$Q_s = (2eI_c C_{nw}/\hbar)^{1/2} R_{qp}$	$Q_n = (2e/h)(2\Lambda_Q/L)V_s R_{ps} C_{nw}$
Potential barrier height	$\Delta U(I) = (hI_c/2\pi e)[(1-(I/I_c)^2)^{1/2} - (I/I_c)\arccos(I/I_c)]$	-

This more detailed description of the electrodynamics of YBCO phase-slip nanowires in terms of the extended RCSJ model is now included in Supplementary Information.

II. If their analysis is correct in some way, then very likely the nanowire contains a Josephson junction that dominates the transport. In this case, they observed MQT of YBCO Josephson junction, not phase slip nanowire.

Reply: We appreciate the reviewer's concern and take this opportunity to assess the transport behavior of our nanowires. We find transport properties that are in stark disagreement with the reviewer's hypothesis of our nanowires' containing 1D weak links that would result in behavior characteristic of Josephson junctions:

1. The YBCO phase-slip nanowires show all features characteristic of phase-slip processes: (i) we observe direct voltage switching from the superconducting to the resistive state, (ii) we find that the differential resistance of each resistive branch is proportional to the branch number, (iii) we find that the resistive branches have nearly the same excess current, and (iv) we could show that the the voltage switching behavior is

well-described by the phases-slip model ^{7,15,16}. This behavior is inconsistent with that of a Josephson junction as hypothesized by the reviewer.

2. Phase-slip dynamics in wide ($W > 4.4\xi$) bridges are only observable if their line-edge roughness is very small ¹. The fact that we observe IV curves characteristic of phase-slip processes thus excludes the existence of constrictions in the nanowires that could act as a one-dimensional weak links, providing further evidence that Josephson junctions do not dominate the transport properties of our nanowires.

3. The voltage switching with current hysteresis due to the phase slippage cannot be observed for short superconducting constrictions¹⁴.

The above points were discussed extensively in our previous work and we refer the reviewer to Ref. [19] in the main text.

We have included further information on the phase-slip processes in the main text to clarify this point.

III. If they assume a JJ with 120 Ohm (from fig1c, it looks more like 20 Ohm) normal resistance and several fF capacitance, which is typical for JJs of their size, they will have plasma frequency of what they observed.

Reply: We note that the dc current-voltage characteristic of both Josephson junctions and phase-slip nanowires is a result of time averaged voltage oscillations where superconducting and normal components carry the current intermittently. In this case, the (normal) resistance generally cannot be calculated as $R = V/I$, where I is the total current through the nanowire. In our manuscript, we follow the *Skocpol–Beasley–Tinkham* model of the phase-slip process in a superconducting nanowire ^{7,15} to calculate the resistance of the nanowire in the normal state (i.e. the resistance of the phase-slip line in our case) as $R_d = dV/dI = 120 \Omega$.

If the nanowire contains a grain-boundary that acts as a weak link (as it is assumed by the reviewer), an intrinsic capacitance of such grain-boundary across the 55-nm-wide and 5-nm-thick YBCO nanowire can be estimated as $C_{gb} = \epsilon_0 \epsilon (Wd/t) = 0.005-0.02$ fF, where ϵ_0 is the vacuum permittivity, $\epsilon = 5-7$ is the dielectric constant of the YBCO grain-boundary, W is the nanowire width, d is the nanowire thickness, and $t = 1-2$ nm is the grain-boundary thickness. The contribution of the SrTiO₃ substrate can be estimated as $C_{STO} \approx 0.7$ fF ¹⁷. Here, the total capacitance will not exceed 0.8-0.9 fF, which is 5 times smaller than the capacitance used in our

calculation of the plasma frequency. The capacitance of a weak link with non-tunneling conductivity (e.g. a constriction or part of the nanowire with reduced superconducting properties) would be even smaller. The observed plasma frequencies therefore disagree with those one would expect for weak links in the nanowire.

IV. On the other hand, MQT (not quantized energy level) in high T_c oxide material in Josephson regime was first demonstrated by Inomata et al PRL 95, 107005 (2005), but they fail to mention it.

Reply: We were aware of this work but had omitted it in the interest of a concise introduction. We thank the reviewer for reminding us of the importance of Inomata's work and have now included references to MQT and ELQ in high- T_c Josephson junctions in the "Introduction".

V. Concerning the claimed 1.5 ms lifetime of the excited states, I am not so sure where did they get the value $R_{qp} > 1.5 \times 10^{10}$ Ohm in order to obtain the lifetime. From figure 1c, the resistance is near 20 Ohm.

Reply: As discussed above (see our reply to comment I), the resistance that leads to dissipation in the nanowire is different in the superconducting and resistive states. In the superconducting state dissipation occurs due to thermally equilibrium quasiparticles (thermal excitations from the superconducting ground state ⁷), which are exponentially small in number at temperatures well below the critical temperature. In the resistive state dissipation is determined by the resistance of the phase-slip line. This modification of the RCSJ model is similar to that used to describe the electrostatics of the unshunted tunnel Josephson junctions (RSJN model) where the "leakage" resistance at $V < \Delta/e$ is significantly higher than the normal-state resistance at voltages $V > \Delta/e$ ⁶. We estimate the quasiparticle resistance by employing the theory developed by Sivestrini, Ovchinnikov and Cristiano (SOC), which yields $R_{qp} = 3 \cdot 10^{11} \Omega$ and a corresponding lifetime in the excited state of 1.5 ms at $T = 14-16$ K. Sivestrini, Ovchinnikov, and Cristiano consider the shape-dependence of the SCD for different bias current ramp rates, dI/dt , and predict that the lower energy levels become accessible in the switching-current measurement and nearly equally spaced peaks appear in the SCD at temperatures slightly above T_c and high (non-adiabatic) dI/dt rates ¹⁹. The predictions of the SOC model were experimentally tested in Ref. [18, 20,21]. Figure 1 (labeled FIG. 2 in reference [19]) shows an example of the transformation of the switching-current distribution with increasing current ramp rate for a low- T_c Josephson junction, where

[Redacted]

Figure 1. Dependence of the switching-current distribution of the Nb-AlO_x-Nb Josephson junction on the current ramp rate at different temperatures. Figure is taken from Ref. [18]

energy level quantization was confirmed, similar to several other studies. The escape from lower energy levels is observable if the following expression is fulfilled

$$[(dI/dt)R(T)C/I_c]^{-1} < 200 \quad (3)$$

where dI/dt is the bias current rate, I_c is the critical current, R is the quasiparticle resistance, and C is the capacitance. We observe a similar transformation of the switching-current distribution for our nanowire (Figure 2a in the manuscript and Figure S1 in the Supplementary material) and attribute it to the presence of quantized energy levels. We would like to emphasize that the SOC model is the only known model, which can explain the temperature dependent variation of the

SCD observed in our experiments. From inequality (3) we estimate the quasiparticle resistance of the nanowire as $R_{qp} > 3 \cdot 10^{11} \Omega$. We would like to point out that this is a relatively rough order-of-magnitude estimate of the quasiparticle resistance because the oscillations of the switching-current distribution presented in our work are not as clear as those observed in Ref [18] for a low- T_c Josephson junction. However, we can compare the quasiparticle resistance obtained from the SOC model with the quasiparticle resistance one would expect for quasiparticles in thermal equilibrium with the bath and $\Delta = 25$ meV. Then the quasiparticle resistance at $T = 16$ K can be calculated as $R_{qp}(T=16K) = R_n \exp(\Delta/kT) = 3 \cdot 10^{11} \Omega$ that is in agreement with $R_{qp} > 3 \cdot 10^{11} \Omega$ obtained from inequality (3). Here $R_n = 4200 \Omega$ is the nanowire resistance in the normal state. We have revised the description of the SOC model in the main text to explain the calculation of R_{qp} in more detail.

VI. *The experimentally obtained line width in fig 2a gives a much shorter time constant.*

We would like to clarify that the data presented in fig. 2a in the manuscript are the switching-current distributions rather than the energy level line widths. The switching-current distributions presented in the manuscript represent the probabilities for a nanowire in the ground or excited energy state to switch into the resistive state as the bias current increases. We interpret these data via the mechanical analog of the RCSJ model, i.e. movement of a particle in a tilted washboard potential. In the absence of fluctuations, the particle escapes a potential well when the potential barrier ΔU is zero, as shown in figure 2b, which occurs at some current referred to as zero-fluctuation critical current I_c . However, real underdamped junctions switch into the resistive state at currents $I_s < I_c$ when the potential barrier $\Delta U > 0$ (figure 2a) due to thermal or quantum fluctuations. Within the framework of the RCSJ model the potential barrier height is given by the expression $\Delta U(I) = (hI_c/2\pi e)[(1-(I/I_c)^2)^{1/2} - (I/I_c)\arccos(I/I_c)] \approx hI_c/\pi e(1-I/I_c)^{3/2}$ and thermally activated (TA) escape from the potential well occurs at a rate of $\Gamma_{TA} = a_T(\omega_p/2\pi)\exp[-\Delta U/kT]$ ²². If the temperature is low enough, however, the particle escapes the well by the tunneling through the potential barrier (referred to macroscopic quantum tunneling (MQT)). Here the escape rate from the potential well by MQT is given by $\Gamma_{MQT} = a_q(\omega_p/2\pi)\exp[(-36\Delta U/5\hbar\omega_p)(1+0.87/Q)]$ ²². The temperature where $\Gamma_{TA} = \Gamma_{MQT}$ is called the crossover temperature between TA and MQT regimes. The exponential dependence of the escape rate on the potential barrier height results in the characteristic asymmetric switching-current distribution that can be calculated as²³

Figure 2. Escape of the particle from the tilted washboard potential at different bias currents.

$$P(I) = \Gamma \left(\frac{dI}{dt} \right)^{-1} \left(1 - \int_0^I P(u) du \right)$$

As we observe the escape from the different quantum levels (figure 3 in the manuscript) at relatively slow bias current sweep rates, it is reasonable to expect $Q \gg 1$. Then the expression for the quantum tunneling escape rate reduces to $\Gamma_{\text{MQT}} = a_q(\omega_p/2\pi)\exp[(-36\Delta U/5\hbar\omega_p)]$, which does not depend on the quality factor anymore. Hence, it is not possible to estimate the lifetime of the exited state from the width of the measured SCDs.

We have added an explanation of the physical picture behind SCDs in the section “Experimental results and discussion” of the main text.

The revisions made in response to the reviewer’s comments are in the summary of changes and highlighted in the manuscript.

References

- 1 Andronov, A., Gordion, I., Kurin, V., Nefedov, I. & Shereshevsky, I. Kinematic Vortices and Phase-Slip Lines in the Dynamics of the Resistive State of Narrow Superconductive Thin-Film Channels. *Physica C* **213**, 193-199, (1993).
- 2 Sivakov, A. G. *et al.* Josephson behavior of phase-slip lines in wide superconducting strips. *Physical Review Letters* **91**, 267001 (2003).

- 3 Likharev, K. K. Superconducting Weak Links. *Reviews of Modern Physics* **51**, 101-159, (1979).
- 4 Mccumber, D. E. Effect of Ac Impedance on Dc Voltage-Current Characteristics of Superconductor Weak-Link Junctions. *J Appl Phys* **39**, 3113-3118, (1968).
- 5 Stewart, W. C. Current-Voltage Characteristics of Josephson Junctions. *Appl Phys Lett* **12**, 277, (1968).
- 6 Likharev, K. K. *Dynamics of Josephson Junctions and Circuits*. (Gordon and Breach, 1986).
- 7 Tinkham, M. *Introduction to superconductivity*. (McGraw-Hill, Inc., 1996).
- 8 Matveev, K. A., Larkin, A. I. & Glazman, L. I. Persistent current in superconducting nanorings. *Physical Review Letters* **89**, 096802 (2002).
- 9 Khlebnikov, S. Decay of near-critical currents in superconducting nanowires. *Physical Review B* **94**, 064517 (2016).
- 10 Golubov, A. A., Kupriyanov, M. Y. & Il'ichev, E. The current-phase relation in Josephson junctions. *Reviews of Modern Physics* **76**, 411-469 (2004).
- 11 Arutyunov, K. Y., Hongisto, T. T., Lehtinen, J. S., Leino, L. I. & Vasiliev, A. L. Quantum phase slip phenomenon in ultra-narrow superconducting nanorings. *Sci Rep-Uk* **2**, 293 (2012).
- 12 Arpaia, R. *et al.* Ultra low noise $\text{YBa}_2\text{Cu}_3\text{O}_{7-\delta}$ nano superconducting quantum interference devices implementing nanowires. *Appl Phys Lett* **104**, 072603 (2014).
- 13 Kautz, R. L. & Martinis, J. M. Noise-Affected IV Curves in Small Hysteretic Josephson-Junctions. *Physical Review B* **42**, 9903-9937, (1990).
- 14 Michotte, S., Matefi-Tempfli, S., Piraux, L., Vodolazov, D. Y. & Peeters, F. M. Condition for the occurrence of phase slip centers in superconducting nanowires under applied current or voltage. *Physical Review B* **69**, 094512 (2004).
- 15 Skocpol, W. J., Beasley, M. R. & Tinkham, M. Phase-Slip Centers and Nonequilibrium Processes in Superconducting Tin Microbridges. *J Low Temp Phys* **16**, 145-167, (1974).
- 16 Tinkham, M. Interaction of Phase-Slip Centers in Superconducting Filaments. *J Low Temp Phys* **35**, 147-151, (1979).

- 17 Lyatti, M., Poppe, U., Gundareva, I. & Dunin-Borkowski, R. High-resistance $\text{YBa}_2\text{Cu}_3\text{O}_{7-x}$ grain-boundary Josephson junctions fabricated by electromigration. arxiv.org/abs/1908.03784 (2019)
- 18 Silvestrini, P., Palmieri, V. G., Ruggiero, B. & Russo, M. Observation of energy levels quantization in underdamped Josephson junctions above the classical-quantum regime crossover temperature. *Physical Review Letters* **79**, 3046-3049, (1997).
- 19 Silvestrini, P., Ovchinnikov, Y. N. & Cristiano, R. Effects of Level Quantization on the Supercurrent Decay in Josephson-Junctions - the Nonstationary Case. *Physical Review B* **41**, 7341-7344, (1990).
- 20 Ruggiero, B. *et al.* Effects of energy-level quantization on the supercurrent decay of Josephson junctions. *Physical Review B* **59**, 177-180, (1999).
- 21 Yamanashi, Y., Ito, M., Tagami, A. & Yoshikawa, N. Observation of quantized energy levels in a Josephson junction using SFQ circuits. *Ieee Transactions on Applied Superconductivity* **15**, 864-867, (2005).
- 22 Garg, A. Escape-Field Distribution for Escape from a Metastable Potential Well Subject to a Steadily Increasing Bias Field. *Physical Review B* **51**, 15592-15595, (1995).
- 23 Fulton, T. A. & Dunkleberger, L. N. Lifetime of Zero-Voltage State in Josephson Tunnel-Junctions. *Physical Review B* **9**, 4760-4768, (1974).

Response to reviewer 3

The authors would like to thank the reviewer for his/her valuable comments, which helped to improve the manuscript. Our response follows below (the reviewer's comments are given in italics).

The main advance on previous light/induced phase slip experiments appears to be that single photons are used to change state, and the resulting resistance change is measured statistically. Without doubt the paper is potentially interesting to readers of Nature communications, particularly the realization of single photon detection using SC nanowires.

- I. *However, I am not convinced about the applicability of the RCSJ model to the present experiments - or at least the way it is presented.*

Reply: Reviewer 2 raised a similar concern, which we replied to in much detail. We here refer the reviewer to our reply to comment I of reviewer 2.

The detailed description of the electrodynamics of YBCO phase-slip nanowires in terms of the extended RCSJ model is now included in Supplementary Information.

- II. *I am concerned about the interpretation (e.g. Fig. 2) in terms of quantised levels.*

Reply: We trust that our detail explanation of the applicability of the RCSJ model and the good quantitative agreement of all our experimental data with the model predictions (including energy level quantization) provide very convincing evidence for the observation of energy level quantization in the YBCO nanowires.

To the best of our knowledge, there is only one model that predicts oscillations of the switching-current distribution of the Josephson junction that occur at temperatures slightly above the TAMQT crossover temperature and disappear at higher and lower temperature. This model, introduced by Silvestrini, Ovchinnikov, and Cristiano (SOC), considers the dependence of the switching-current distribution (SCD) on the bias current ramp rate ¹. The model predicts that the decay from the lower energy levels becomes accessible at the temperatures slightly above the crossover temperature if the bias current ramp rate is high enough, i.e. non-adiabatic. The

predictions of the SOC model were experimentally verified in Ref. [2-4]. Here we use the SOC model to estimate the quasiparticle resistance.

We find good agreement of our experimental results with the SOC model in two notable aspects: Firstly, figure 4b clearly shows the transition from the “adiabatic” to “non-adiabatic” bias current ramp rate, when lower energy levels become accessible. Secondly, we compare the quasiparticle resistance obtained within the framework of the SOC model ($R_{qp}(T \approx 14-16K) = 3 \cdot 10^{11} \Omega$) with the quasiparticle resistance one would expect for thermal equilibrium with the bath and $\Delta = 25$ meV. For the latter the quasiparticle resistance at $T = 16$ K can be calculated as $R_{qp}(T=16K) = R_n \exp(\Delta/kT) = 3 \cdot 10^{11} \Omega$, which is in agreement with $R_{qp} \approx 3 \cdot 10^{11} \Omega$ obtained within the SOC model. Here $R_n = 4200 \Omega$ is the nanowire resistance in the normal state.

On the contrary, if the broadening and variation of the switching current distribution was caused by frequency-modulated extrinsic noise ⁵, then the broadening and variation of the switching-current distribution should not depend on temperature and the retrapping-current distribution would have to be broadened and modulated as well ⁶. None of these effects were observed in our experiments, thus excluding extrinsic noise as a detrimental influence on our measurements.

We have now added further information on the use of the SOC model to the main text.

III. *How is it justified to consider that a 50 nm wide and 5 nm thick YBCO strip is a quasi-1D nanowire? The in-plane coherence length in this material is only 2 nm in plane, and <1 nm out of plane? Are there multiple channels? How do they function?*

Reply: According to both, the theoretical description of superconducting weak links ⁷ and corresponding numerical simulations ⁸, 1D phase-slip centers described within the *Skocpol–Beasley–Tinkham* model ⁹ can appear in superconducting nanowires with width $W \leq 4.4\xi$, where ξ denotes the coherence length. However, the ladder-like current-voltage characteristic of phase-slip processes were observed also for wider nanowires $W \gg \xi$ ¹⁰. In the wide nanowires phase slippage occurs either by the 2D analog of a phase-slip center (i.e. the order parameter is suppressed across the entire nanowire) or a so-called vortex-street (i.e. running phase-slip centers, which are also referred to as kinematic vortices) ^{11,12}. For our nanowires with width $W > 40\xi$ we consider the vortex-street mechanism as energetically favorable. In the manuscript we will refer to the phase-slip process in our YBCO nanowires using the well-established term

"phase-slip line", bearing in mind that the phase slippage occurs due to the motion of kinematic vortices.

We have now added a more detailed explanation of the phase slippage in the YBCO nanowires in the section "YBCO phase-slip nanowires" of the main text.

IV. *If this is a SC nanowire, then one should observed steps in the CVCs corresponding to different configurations of PSC along the wire. Why are they not observed?*

Reply: As the reviewer correctly noticed, in figure 1c we show only the first voltage switching event of the CVCs such that the (first) switching and retrapping currents are clearly discernible. For 2- μm -long nanowires we observe 3 steps in the CVCs corresponding to the different configurations of PSC along the wire as well as the following other features characteristic of phase-slip processes: (i) the differential resistance of each resistive branch is proportional to the branch number, (ii) the resistive branches have nearly the same excess current, (iii) the instances of the voltage switching events are in good qualitative agreement with the model of weakly interacting phase-slip centers^{9,13}. For a more detailed analysis of the voltage switching behavior in these YBCO nanowires we refer the reviewer to our previous work cited as Ref. [19] in the main text.

We have added another reference on our previous work on voltage switching mechanisms in these YBCO nanowires to the sentence "All of the nanowires showed current-voltage (IV) curves that were characteristic to phase slippage." in the main text.

V. *Are the sidebands due to this perhaps?*

Reply: This is an interesting question. Let us consider phase-slip lines that occur stochastically at different positions along the nanowire. It is then reasonable to assume that the relative difference in the switching currents $\Delta I_s/I_s$ will correspond to variations in nanowire width (at the least). For our nanowires we determine the ratio of YBCO lattice parameters in a- and b-directions (≈ 0.39 nm) to the nanowire width as $a/W > 7.1 \cdot 10^{-3}$. However, we find switching current variations of only $\Delta I_s/I_s \approx 5.5 \cdot 10^{-4}$, i.e. much smaller than the a/W -ratio. Therefore, we conclude that it is rather unlikely that the side peaks originate from phase-slips occurring in different locations along the nanowire.

On the other hand, one may presume that the side peaks appear due to the interaction of the Josephson plasma mode with another resonant system. Possible candidates for the latter are geometric resonances or other plasma modes, i.e. different from Josephson plasma modes, with smaller energy level spacing. We believe that the temperature dependence may be an important property to elucidate the origin of the side peaks in future studies, which are however beyond the scope of the work presented here.

VI. *The temperature dependence also doesn't make sense to me: within a quantised level picture, if the current is ramped, then above a threshold value, switching to a resistive state would be expected at the ground state value (which is always the lowest, at any temperature). Why is the ground state shifting with T? To observe a T-dependent shift of the current, the shunt resistance would need to be temperature dependent, which would change the potential also.*

Reply: We would like to clarify that the energy level quantization of our YBCO nanowires refers to the energy-phase space (in our nanowires phase is a “good” quantum variable) rather than the energy-current space, as the reviewer may have assumed. Hence, the presented switching-current distributions (SCD) should not be interpreted as a direct representation of energy levels, e.g. including the ground state.

To understand the relation between the measured SCDs and the energy levels in the nanowire it is instructive to consider the nanowire (shunted by the capacitance C) as an LC circuit with resonance frequency $\omega_p = 1/(LC)^{1/2}$, where L is the inductance of the nanowire. Within the framework of the RCSJ model the inductance of the nanowire is given as

$$L = \Phi_0/2\pi I_c(T)\cos\varphi = L_{j0}[1-(I/I_c)^2]^{1/2} \quad (1),$$

where $L_{j0} = \Phi_0/2\pi I_c(T)$ is the zero-bias Josephson inductance. Using equation (1), the zero-bias resonance frequency of the LC circuit (Josephson plasma frequency) can be calculated as $\omega_{p0} = (2eI_c/\hbar C)$. It is clear from equation (1) that the inductance, the resonant frequency, and, hence, the energy of the ground state depend on the critical current of the nanowire and on the bias current. Note that the resonance frequency is temperature dependent due to the temperature dependence of the critical current $I_c(T)$, as also observed in the inset of Figure 1c. Consequently the energy of the ground state, $E_0 = \hbar\omega_p/2$, must also depend on temperature and bias current.

The SCDs presented in the manuscript on the other hand represent the probability of a nanowire in the ground or excited state to switch into the resistive state.

If the zero-fluctuation critical current I_c is temperature independent (as it is the case in low- T_c superconductors at $T \ll T_c$) the position of the peak in the SCD shifts towards the zero-fluctuation critical current I_c in the TA regime and is temperature independent in the MQT regime. In contrast to low- T_c superconductors, the critical current of our YBCO nanowires scales nearly linear with temperature even at temperatures close to zero, therefore, the potential barrier height and, hence, the position of the SCD peak are temperature dependent in both TA and MQT regimes. Please see our reply to comment VI of reviewer 2 for a more detailed description within the RCSJ model.

We have added an explanation of the physical picture behind SCDs in the section “Experimental results and discussion” of the main text.

- VII. *Another issue is the discussion of the photon excitation process. The discussion of the subsequent quasiparticle dynamics is based on the assumption that QP diffusion is dominant. This is not true in high- T_c superconductors, where phonon escape determines the QP lifetime, according to the Rothwarf-Taylor mechanism, particularly in thin films. This has been discussed in the literature quite extensively for YBCO.*

Reply: We thank the reviewer for raising this interesting point. Firstly, we restrict our consideration by the hotspot expansion during the quasiparticle thermalization process with time constant $\tau_{th} = 0.56$ ps, which is shorter than the electron-photon relaxation time τ_{ep} .^{14,15} Hence, we can neglect the energy transfer to phonons during this time. Further, the electron-electron scattering time, $\tau_{e-e} \leq 0.1$ ps, is much shorter than the thermalization time^{14,16} and hence, the quasiparticle motion during the thermalization time is diffusive rather than ballistic¹⁶. Secondly, note that the experimental data reported in the literature for nonequilibrium quasiparticle lifetimes in YBCO ranges over at least two orders of magnitude: the shorter recombination times were found for thin films deposited on substrates that have large lattice constant mismatch with YBCO, where high concentrations of defects are expected. On the other hand, longer recombination times were found for high-purity single-crystal samples with good oxygen ordering^{14,16-19}. One may conclude that scattering on defects has a larger effect on the quasiparticle lifetime than the electron-photon relaxation time τ_{ep} .

Our YBCO thin films show very large critical current density and we hence expect that the relaxation times of our films are closer to the ones for single-crystal samples where $\tau_{\text{th}} \ll \tau_{\text{ep}}$, which is consistent with low concentration of defects. Therefore, we believe that our nanowires are in a strong phonon bottleneck regime, as it was also found with other cuprate superconductors²⁰ and the refined hotspot model introduced by Semenov et al.²¹ is applicable.

We have added a corresponding reasoning concerning nonequilibrium quasiparticle diffusion at $t \leq \tau_{\text{th}}$ to the main text.

In summary, the paper as it stands cannot be recommended. However, I don't think it's beyond repair.

Reply: We trust that the revisions made in response to the reviewer's comments improved the clarity of the manuscript and he/she agrees that all parts that needed repair are in good shape now.

VIII. *The title might want to mention single photon control of phase slips.*

Reply: We are grateful to the referee for this suggestion for the title. The title was revised as "Energy-level quantization and single-photon control of phase slips in $\text{YBa}_2\text{Cu}_3\text{O}_{7-x}$ nanowires"

IX. *In the abstract it is emphasised that the improvement for using YBCO nanowires seemingly have a longer lifetime and remain in the quantum regime at higher temperature than low- T_c nanowires. This is not particularly surprising, considering the larger gap and resulting change in QP recombination dynamics, and distracts the reader from the main issues.*

Reply: We thank the reviewer for emphasizing that our findings are not only very interesting in terms of the observed energy level quantization and long-lived quantum states but also because of the control of phase slips at the single-photon level. We consider the latter aspect equally important as the observation of a response to single-photons in YBCO nanowires opens up very exciting prospects for realizing superconducting nanowire single-photon detectors (SNSPDs) at elevated operating temperatures, which are attractive devices both for novel technological applications as well as for gaining new insight into the detection mechanism of SNSPDs. Nevertheless we reiterate that the observation of energy-level quantization and long-lived excited

state paves the way for nanowire qubits (note that a current-biased YBCO nanowire can be employed as a phase qubit!) and other superconducting quantum circuits with the long decoherence time and the high operating temperature.

The quantum behavior and long lifetimes at higher temperatures may not seem “particularly surprising” to the reviewer, but we have to admit that it came as quite a surprise to us! Also, we do not believe that our experiments would have been any less difficult to realize even if we had expected such behavior. We believe that many readers of Nature Communications will be as surprised and excited by our findings as we were because YBCO is d-wave superconductor with zero superconducting energy gap in the nodal direction. One may therefore be lead to expect that YBCO devices contain a large number of the thermally equilibrium quasiparticles even at temperatures close to zero. This should significantly reduce the excited state lifetime as compared to corresponding low- T_c devices. We hence found it quite surprising to observe quantized, long-lived energy levels in a nanowire fabricated from a d-wave superconductor. We would also like to point out that, to the best of our knowledge, energy-level quantization has been never observed for both low- T_c and high- T_c superconducting nanowires with a finite critical current.

X. *The title might want to mention single photon control of phase slips, which is after all the most interesting aspect of the data.*

The title was changed to “Energy-level quantization and single-photon control of phase slips in $\text{YBa}_2\text{Cu}_3\text{O}_{7-x}$ nanowires”.

The revisions made in response to the reviewer’s comments are in the summary of changes and highlighted in the manuscript.

References

- 1 Silvestrini, P., Ovchinnikov, Y. N. & Cristiano, R. Effects of Level Quantization on the Supercurrent Decay in Josephson-Junctions - the Nonstationary Case. *Physical Review B* **41**, 7341-7344, (1990).
- 2 Silvestrini, P., Palmieri, V. G., Ruggiero, B. & Russo, M. Observation of energy levels quantization in underdamped Josephson junctions above the classical-quantum regime crossover temperature. *Physical Review Letters* **79**, 3046-3049, (1997).

- 3 Ruggiero, B. *et al.* Effects of energy-level quantization on the supercurrent decay of Josephson junctions. *Physical Review B* **59**, 177-180, (1999).
- 4 Yamanashi, Y., Ito, M., Tagami, A. & Yoshikawa, N. Observation of quantized energy levels in a Josephson junction using SFQ circuits. *Ieee Transactions on Applied Superconductivity* **15**, 864-867, (2005).
- 5 Fulton, T. A. & Dunkleberger, L. N. Lifetime of Zero-Voltage State in Josephson Tunnel-Junctions. *Physical Review B* **9**, 4760-4768, (1974).
- 6 Tinkham, M. *Introduction to superconductivity*. (McGraw-Hill, Inc., 1996).
- 7 Likharev, K. K. Superconducting Weak Links. *Reviews of Modern Physics* **51**, 101-159, (1979).
- 8 Qiu, C. Y. & Qian, T. Z. Numerical study of the phase slip in two-dimensional superconducting strips. *Physical Review B* **77**, 174517 (2008).
- 9 Skocpol, W. J., Beasley, M. R. & Tinkham, M. Phase-Slip Centers and Nonequilibrium Processes in Superconducting Tin Microbridges. *J Low Temp Phys* **16**, 145-167, (1974).
- 10 Sivakov, A. G. *et al.* Josephson behavior of phase-slip lines in wide superconducting strips. *Physical Review Letters* **91**, 267001 (2003).
- 11 Weber, A. & Kramer, L. Dissipative States in a Current-Carrying Superconducting Film. *J Low Temp Phys* **84**, 289-299, (1991).
- 12 Andronov, A., Gordion, I., Kurin, V., Nefedov, I. & Shereshevsky, I. Kinematic Vortices and Phase-Slip Lines in the Dynamics of the Resistive State of Narrow Superconductive Thin-Film Channels. *Physica C* **213**, 193-199, (1993).
- 13 Tinkham, M. Interaction of Phase-Slip Centers in Superconducting Filaments. *J Low Temp Phys* **35**, 147-151, (1979).
- 14 Sobolewski, R. Quasiparticle thermalization and recombination in high-temperature superconductors excited by femtosecond optical pulses. *Lect Notes Phys* **545**, 100-122, (2000).
- 15 Semenov, A. D., Gol'tsman, G. N. & Sobolewski, R. Hot-electron effect in superconductors and its applications for radiation sensors. *Supercond Sci Tech* **15**, R1-R16, (2002).

- 16 Gedik, N., Orenstein, J., Liang, R. X., Bonn, D. A. & Hardy, W. N. Diffusion of nonequilibrium quasi-particles in a cuprate superconductor. *Science* **300**, 1410-1412, (2003).
- 17 Gedik, N. *et al.* Single-quasiparticle stability and quasiparticle-pair decay in $\text{YBa}_2\text{Cu}_3\text{O}_{6.5}$. *Physical Review B* **70**, 014504 (2004).
- 18 Averitt, R. D. *et al.* Nonequilibrium superconductivity and quasiparticle dynamics in $\text{YBa}_2\text{Cu}_3\text{O}_{7-\delta}$. *Physical Review B* **63**, 140502(R) (2001).
- 19 Luo, C. W. *et al.* Spatial symmetry of the superconducting gap of $\text{YBa}_2\text{Cu}_3\text{O}_{7-\delta}$ obtained from femtosecond spectroscopy. *Physical Review B* **68**, 220508(R) (2003).
- 20 Kabanov, V. V., Demsar, J. & Mihailovic, D. Kinetics of a superconductor excited with a femtosecond optical pulse. *Physical Review Letters* **95**, 147002 (2005).
- 21 Semenov, A., Engel, A., Hubers, H. W., Il'in, K. & Siegel, M. Spectral cut-off in the efficiency of the resistive state formation caused by absorption of a single-photon in current-carrying superconducting nano-strips. *Eur Phys J B* **47**, 495-501, (2005).

Reviewers' comments:

Reviewer #1 (Remarks to the Author):

The authors have addressed all my remarks in the new version of the manuscript. I think now it is ready to be published.

Reviewer #2 (Remarks to the Author):

The authors have answered my comments and concerns systematically in their reply. Now I think the analysis of the obtained data based on RSJ model can be justified.

Reviewer #3 (Remarks to the Author):

This is my second review of the paper by Lyatti et al.

The authors have addressed some of the minor issues that were raised in my first report, which I find satisfactory.

As mentioned in my first report, the major novelty - if proven - would be single photon control of PSN which would be an advance worth reporting. I was not convinced about this, and am even less convinced now. To claim single photon control, the photon flux should be sparse, so that individual events could be observed and statistics accumulated in this way. The LED source emits a continuous flux of a huge number of photons. Single photon nanowire detectors are quite common these days, and the literature shows how to demonstrate the effect.

Other issues:

I still find the discussion of the processes following photon absorption by YBCO too superficial and misleading. The authors are apparently not aware of the vast literature on the subject. The arguments and the numbers are in fact wrong and incorrectly interpreted: the thermalization time has been measured by many authors directly, e.g. Gadermaier, et al D. PRX 4, 011056 (2014) and analyzed in terms of the Allen model (Phys Rev Lett 59, 1460-1463 (1987)), or more recent by Kabanov and Alexandrov model (Phys Rev B 78, 174514 (2008)). On the other hand, the quasiparticle relaxation dynamics in YBCO has been analysed in terms of Rothwarf-Taylor dynamics and the values are about one order of magnitude shorter than the value quoted. In fact, the dynamics will be governed by other factors which the authors don't discuss.

The RCSJ model which they use has been extensively elaborated by Bezryadin in the last 10 years whom they don't cite for some reason. Similar statistics as they present here have been shown before. The energy level quantisation as an alternative to describing nanowire dynamics can be found in the literature as well, so it is not new. Since many of the phenomena they discuss have already been presented previously, it would help the reader if the authors referred to them.

Unfortunately, I still find the paper very hard to follow. The response to reviewers hasn't made it better.

In summary, the main claim of single-photon control is not supported by evidence. Energy level

quantisation, which also appears in the title is not particularly new.
As a result, I cannot recommend the paper for publication in Nature Communications.

Second response to the reviewer 3.

The authors would like to thank the reviewer for his/her critical assessment of our work. Our response follows below (the reviewer's comments are given in italics).

This is my second review of the paper by Lyatti et al.

The authors have addressed some of the minor issues that were raised in my first report, which I find satisfactory.

I. As mentioned in my first report, the major novelty - if proven - would be single photon control of PSN which would be an advance worth reporting.

Reply: We agree with the reviewer that single photon control of phase slip nanowires (PSNs) is one major novelty of our work. However, we would like to point out that the observation of quantized energy levels (ELQ) in nanowires made from a high-temperature superconductor with finite critical current is by no means less important, as also recognized by the other two reviewers. The observed ELQ will certainly be of very high relevance to the large community interested in quantum computing with superconducting qubits. To clarify the context in which ELQ displays its compelling relevance, we here provide another perspective on our work, which should highlight this point.

Quantum computing is one of the most recognized developments in current science¹. Many of the most advanced physical implementations of quantum computing tasks rely on superconducting circuits, where Josephson junctions provide the nonlinearity required for selective access to quantum levels. For technological reasons these circuits are currently realized (predominantly) with aluminum tunnel Josephson Junctions, which require operating temperatures of a few tens of mK. Quantum computers featuring more than 50 such qubits made from low-temperature superconductors already exist², however their long term perspective remains unclear because they have moderate quality factors of no more than a few million^{3,4}. Despite the current success with these systems it is of utmost importance to find new ways to very significantly improve on the device quality factors, which limit the number of quantum operations that can be performed with a superconducting circuit before information is lost due to decoherence. In our work we demonstrate energy level quantization in superconducting YBCO nanowires, suitable for qubit encoding, for the first time. Our YBCO nanowires show very high quality factors in excess of 10^{10} , thus dramatically improving the prospects of performing very large numbers of quantum gate operations within the coherence time in future

devices. Further, in our YBCO nanowires the transition into the quantum regime occurs at record high temperatures of 12-13 K, which will significantly alleviate the wiring and cooling demands of the overall system. To the best of our knowledge, there neither are previous reports on ELQ in superconducting nanowires with finite critical current, nor could we find crossover temperatures above 2 K for superconducting quantum devices (which was reported for Pb-Pn(In) Josephson junctions ⁵, where excessive loss prevents any useful functionality in the quantum regime). The observation of quantized energy levels, very high quality factors, and record high crossover temperature between classical and quantum regimes for our YBCO nanowires constitute a breakthrough for the development of novel nanowire qubits. We are very confident that the importance of these findings is on a par with our observation of single-photon control of PSNs, which has obvious implications for single-photon detection, as the reviewer already recognized.

II. I was not convinced about this, and am even less convinced now. To claim single photon control, the photon flux should be sparse, so that individual events could be observed and statistics accumulated in this way. The LED source emits a continuous flux of a huge number of photons.

Reply: The reviewer is of course right in that the LED source emits a continuous flux of a relatively large number of photons. The relevant parameter for single-photon control however is the number of photons absorbed in the nanowire. This number is small. Firstly, the nanowire occupies a tiny part of the solid angle into which the LED emits its photons and secondly, only a small fraction of the photons reaching the wire are actually absorbed because the nanowire is ultra-thin. The latter is described by the absorption coefficient k of the YBCO film. Comparing the LED irradiance to the $W = 55$ nm wide and $L = 2 \mu\text{m}$ long nanowire we estimate that no more than $N = \frac{kLWI_{LED}}{E_{ph}} \approx 53,000$ photons/sec are absorbed by the nanowire, where $E_{ph} = 2.7$ eV is the photon energy, $I_{LED} = 0.7$ W/m² is the LED irradiance, and $k \approx 0.3$ is the theoretical limit for the absorption coefficient of a metal film, which we calculate as $k = 4(R_s/Z_0)/[(R_s/Z_0)(n_{sub}+1)+1]^2$, where $n_{sub} = 2.5$ is the index of refraction of the substrate, $R_s = 115 \Omega$ is the surface resistance of the YBCO film measured just above the critical temperature, and $Z_0 = 377$ is the free-space impedance ⁶. Therefore, the absorption rate is one photon per 19 μsec on average. This time interval is six orders of magnitude longer than the quasiparticle recombination time $\tau_r = 1.1 - 10$ ps in optimally doped YBCO thin-films or single-crystal samples ⁷⁻¹². The perturbation caused by a photon will decay long before a subsequent photon will be absorbed by the nanowire. Based on the above, we consider our nanowire to be in the single-photon regime where multiphoton processes have extremely low probability and can be neglected.

In addition, we rule out the effect of a bolometric response on the switching-current statistics because: (i) the observed changes in the switching-current statistics are inconsistent with the bolometric effect, which would allow for a shift of the distribution but could not lead to a change of its shape; (ii) the heat load from optical radiation is smaller than or comparable with the heat load from the 77 K black body radiation from the corresponding radiation shield, $P = \sigma T^4 = 5.67 \cdot 10^{-8} \cdot 77.4^4 = 2 \text{ W/m}^2$; (iii) we did not observe any effect of the optical radiation on the temperature sensor, which is located close to the sample.

We have now included an estimation of the photon flux through the nanowire in the manuscript.

III. Single photon nanowire detectors are quite common these days, and the literature shows how to demonstrate the effect.

Reply: Superconducting nanowire single-photon detectors (SNSPDs) are indeed a rapidly developing technology that enables a plethora of novel experiments at the forefront of several scientific disciplines. All SNSPDs realized to date are made from low temperature superconducting thin-films. These conventional low- T_c SNSPDs are characterized by hotspot creation scenarios and an electro-thermal feedback that result in relatively long recovery times, given either by the time required to cool a hotspot below the critical temperature T_c or by the L_k/R_a -ratio. Here L_k is the kinetic inductance and $R_a = 50\Omega$ is the load impedance (typically of an RF amplifier). Due to the relatively slow recovery, single-photon events detected with low- T_c nanowires can be recorded with high-bandwidth oscilloscopes.

The absorption of single-photons in high- T_c superconducting nanowires, on the other hand, induces a cyclic phase-slip process, which is subject to dynamics on completely different time scales. The heating of the phase-slip nanowire in the resistive state is negligible compared to the critical temperature. Therefore, the recovery time of YBCO nanowires is defined either by the quasiparticle recombination time or by L_k/R_a . Both mechanisms here result in recovery times on the ps-time scale for a 2 μm -long YBCO nanowire. Resolving such short pulses on an oscilloscope would require THz bandwidth, which is not readily available in current technology.

The literature on detecting single-photons with low- T_c nanowires does hence not show how to demonstrate the effect. This is also evident from recognizing that several groups with outstanding material, fabrication, and measurement expertise have unsuccessfully tried to show single-photon detection in high- T_c superconducting devices for more than a decade by relying predominantly on techniques established for low- T_c SNSPDs. We here mastered this challenge by approaching the

problem from a different angle, i.e. measuring the switching-current statistics, which provides an alternative mean of observing the effect of absorbing single-photons in a nanowire made from a high-Tc superconducting material.

Other issues:

IV. I still find the discussion of the processes following photon absorption by YBCO too superficial and misleading. The authors are apparently not aware of the vast literature on the subject.

Reply: In our work we use the “refined hotspot model” introduced by the Semenov et al.¹³ because (i) this model describes the local quasiparticle dynamics caused by a single optical photon and (ii) the description of the processes that occur in the nanowire after photon absorption (see pages 498-499) is identical to the description of processes preceding the phase slip as given by M. Tinkham in “Introduction to superconductivity”¹⁴ (see page 428).

We then follow the description of the energy down conversion process as it is given in the “refined hotspot model” and as commonly done in recent literature. We agree with the reviewer that the description of the energy down conversion process within the framework of the “refined hotspot model” is significantly simplified. In fact, it includes only the third stage of the energy down conversion that takes place in the energy range $\Delta < E < (\text{a few } \Delta)$, where the better part of quasiparticle multiplication occurs due to e-e collisions^{7,15}. This simplification does not affect the general validity of the model because this third stage of the down conversion process lasts much longer than all stages of the preceding cascade combined¹⁵. Details on the approximations used in the “refined hotspot model” can be found in the publication of Semenov et al., which we cite as Ref. [42] in the main text. We find it redundant to include a detailed description of the energy down conversion process in our manuscript, which goes far beyond the framework of the “refined hotspot model”.

The “refine hotspot model” further assumes that during the thermalization process the quasiparticles diffuse out of the photon absorption site. As we mentioned in our previous reply, this assumption is valid for YBCO since the electron-electron scattering time, $\tau_{e-e} \leq 0.1$ ps, is much shorter than the thermalization time $\tau_{th} \approx 0.6$ ps. Within the framework of the “refined hotspot model” the number of quasiparticles at $t = \tau_{th}$, where τ_{th} is the thermalization time, is calculated as $N = \zeta E / \Delta$, where $\zeta \leq 1$ is the efficiency of the quasiparticle multiplication. In our manuscript we assume that the entire photon energy is transferred to the quasiparticles so that $\zeta = 1$. This assumption is valid for $\tau_{th} \gg \tau_r, \tau_{es}$, where τ_r and τ_{es} are the quasiparticle recombination and phonon escape times, respectively¹⁶. The

[Redacted]

Fig.1. Transient reflectivity change of the optimally doped twinned 200-nm-thick YBCO film. Figure is adopted from Ref. [8].

(Phys Rev B 78, 174514 (2008)). On the other hand, the quasiparticle relaxation dynamics in YBCO has been analysed in terms of Rothwarf-Taylor dynamics and the values are about one order of magnitude shorter than the value quoted. In fact, the dynamics will be governed by other factors which the authors don't discuss.

We thank the reviewer for the critical assessment of our findings and would like to clarify that our numbers are in fact consistent with related findings in other reports and correctly interpreted in our manuscript.

To avoid misunderstanding in definitions of the thermalization and quasiparticle recombination times we would like to clarify these terms. Fig. 1 shows typical reflectivity changes after a 20 fs optical pulse using a pump-probe-technique [ref. 8]. The data illustrate the thermalization and recombination processes in an YBCO thin film. The reflectivity change is proportional to the concentration of non-equilibrium quasiparticles created by the optical pulse. The reflectivity transient consists of two parts: a rapid increase of the reflectivity followed by a relatively slow decay back to the initial state. The rapid increase of the reflectivity occurs due to the quasiparticle multiplication during the thermalization. Following the “refined hotspot model” we consider the time interval between the time,

experimentally measured τ_{th} , τ_r and τ_{es} values that confirm this inequality for YBCO are given further below (see the table 1 in our response to comment #V).

We do not find it constructive to repeat further details of the model description in the manuscript, which were extensively discussed in the literature and would divert the readers’ attention from our main findings.

V. The arguments and the numbers are in fact wrong and incorrectly interpreted: the thermalization time has been measured by many authors directly, e.g. Gadermaier, et al D. PRX 4, 011056 (2014) and analyzed in terms of the Allen model (Phys Rev Lett 59, 1460–1463 (1987)), or more recent by Kabanov and Alexandrov model

when the reflectivity starts to increase, and the time, when the reflectivity reaches the maximum, as the thermalization time. It is clear from fig. 1 that the thermalization occurs on time scale of several hundred fs, which is significantly longer than the time resolution of the pump-and-probe technique. The decay of the reflectivity, that takes much longer than the thermalization, occurs due to quasiparticle recombination. The quasiparticle recombination time is typically in the range of 1.1-10 ps depending on the YBCO sample quality. Since $\tau_{th} \ll \tau_r$ for YBCO thin films on SrTiO₃ substrates or single-crystal samples, we neglect the quasiparticle recombination during thermalization process and calculate the number of the non-equilibrium quasiparticles created by an optical photon with the energy E_{ph} as $N = E_{ph}/\Delta$.

The reviewer implies that the relaxation times in YBCO would be one order of magnitude shorter than the values quoted in our manuscript. If this was true the resistances of phase-slip lines that appear due to the quasiparticle diffusion out of the phase-slip site, would have to be significantly lower than the values we measured for our nanowires. Further, note that we chose to use experimentally measured values (all clearly referenced in our manuscript) rather than theoretical estimations, because the relaxation times in thin films depend on many parameters, including strain and defects, which are typically not included in current theoretical models. The shorter relaxation times hypothesized by the reviewer would hence also be in disagreement with those values reported for the relaxation times for YBCO in the superconducting state.

As for the reports mentioned by the reviewer, we would like to point out that Gadermaier et al. (PRX 4, 011056 (2014)) have measured the electron-phonon relaxation time for high- T_c superconductors at room temperature. Measurements performed at two orders of magnitude higher temperatures will obviously lead to significantly faster relaxation times as compared to the superconducting state (i.e. with an energy gap)!

As for the two-temperature and Rothwarf-Taylor models the reviewer is certainly aware that both assume uniform effective temperatures, quasiparticle and phonon concentrations across the entire device. This is certainly not a valid assumption for our experimental conditions and, unsurprisingly, the two-temperature and Rothwarf-Taylor models will hence provide inadequate descriptions of our data. In our manuscript we instead follow a local approach, which was found to be effective for the description of single-photon effects in superconducting nanowires^{13,16,17}. It considers the multiplication and diffusion of hot electrons out of the photon absorption site, rather than energy transfer to phonons, as the main mechanism of electron cooling¹⁰.

To further underline that our data is consistent with reports that are actually related to our work, we list a number of reports below, in addition to those already referenced, which contain experimentally measured thermalization τ_{th} , recombination τ_r and phonon escape τ_{es} times for a range of YBCO samples:

Ref.	Sample	Temp. (K)	τ_{th} (ps)	τ_r (ps)	τ_{es} (ns)
9	YBCO thin film on MgO and SrTiO ₃ substrates	20		2.7	
8	YBCO twinned thin film	14	≈ 0.57	3.4	
10	Detwinned single crystal	4		3-10	
11	YBCO thin film on MgO substrate	4		1.4-2.2	
18	Ortho-II single-crystal	6		50	
7	YBCO thin film on LaAlO ₃ substrate	80	0.56	1.1	few
12	YBCO thin film on SrTiO ₃ substrate	60		4.7	
19	YBCO thin film on sapphire substrate	4			0.5

The data presented in above table shows that for YBCO films on SrTiO₃ substrates or single-crystal samples $\tau_{th} \sim 0.1\tau_r$ and $\tau_{th} \sim 0.001\tau_s$. Note that the cited values of quasiparticle recombination times are measured in pump-and-probe experiments and consequently depend on pump beam intensity²⁰, which is obviously significantly higher ($\sim 10^{11}$ - 10^{12} W/m²) than in our experiment (0.7-8.3 W/m²). We, therefore, expect that the quasiparticle recombination time is actually even longer for our experimental conditions.

We have now included some additional references on relaxation times in YBCO in the manuscript.

VI. The RCSJ model which they use has been extensively elaborated by Bezryadin in the last 10 years whom they don't cite for some reason. Similar statistics as they present here have been shown before.

Reply: We completely share the reviewer's admiration for the contributions made by A. Bezryadin and his co-author S. Khlebnikov, concerning the description of the electrodynamics and switching-current dynamics of superconducting nanowires. That is why we have in fact cited their publications in the main text as Ref. [6,8,17] and in the Supplementary Information as Ref. [6] and [9], which the reviewer seemed to have overlooked.

We would like to emphasize that A Bezryadin and many others working on low- T_c superconducting nanowires, demonstrated the feasibility of macroscopic quantum tunneling (MQT) but not energy-level quantization (ELQ). A corresponding discussion on MQT and ELQ phenomena in low- T_c nanowires is found in the introduction of our manuscript (lines 43-59). Key publications on MQT in low- T_c nanowires are cited as Ref. [7,16-18].

On the other hand, we are very surprised that the reviewer believes similar statistics to the ones we report had been shown previously. Fig. 3a in our manuscript clearly demonstrates that the nanowire accommodates at least four quantized energy levels, which, as far as we know, has never been observed previously. We suspect that the reviewer's comment can only be the result of a misunderstanding and are happy to clarify this issue further if we were pointed to specific references.

VII. The energy level quantisation as an alternative to describing nanowire dynamics can be found in the literature as well, so it is not new. Since many of the phenomena they discuss have already been presented previously, it would help the reader if the authors referred to them.

Reply: We agree with the reviewer that energy level quantization in superconducting nanostructures has raised significant interest in diverse scientific communities. However, we already state clearly in the introduction of our manuscript (lines 40-42) that here we are concerned with quantum effects in superconducting nanowires with finite critical current. We suspect that the reviewer is confusing works on quantum effects in nanowires with coherent quantum phase-slips (where the critical current is suppressed by quantum fluctuations), SNS nanowire junctions with bound Andreev states or semiconducting nanowires with Majorana states, all of which are significantly outside the scope of our manuscript. If the reviewer remains under the impression that predictions or even experimental demonstrations of energy level quantization in superconducting nanowires with finite critical current exist in the relevant literature, we would be grateful if the reviewer could provide us with specific references.

Unfortunately, I still find the paper very hard to follow. The response to reviewers hasn't made it better.

In summary, the main claim of single-photon control is not supported by evidence. Energy level quantisation, which also appears in the title is not particularly new. As a result, I cannot recommend the paper for publication in Nature Communications.

We trust that the revisions made in response to the reviewer's comments and the reasoning on the importance and novelty of energy-level quantization given in this response resolve all remaining misunderstandings and provide convincing reason to recommend our work for publication.

The revisions made in response to the reviewer's comments are in the summary of changes and highlighted in the manuscript.

References

- 1 Quantum Manifesto.
- 2 Arute, F. *et al.* Quantum supremacy using a programmable superconducting processor. *Nature* **574**, 505-510, (2019).
- 3 Devoret, M. H. & Schoelkopf, R. J. Superconducting Circuits for Quantum Information: An Outlook. *Science* **339**, 1169-1174, (2013).
- 4 Kjaergaard, M. *et al.* *Superconducting Qubits: Current State of Play* (2019). Arxiv.org:1905.13641.
- 5 Jackel, L. D. *et al.* Decay of the Zero-Voltage State in Small-Area, High-Current-Density Josephson-Junctions. *Physical Review Letters* **47**, 697-700, (1981).
- 6 Gol'tsman, G. N. *et al.* Picosecond superconducting single-photon optical detector. *Appl Phys Lett* **79**, 705-707, (2001).
- 7 Sobolewski, R. Quasiparticle thermalization and recombination in high-temperature superconductors excited by femtosecond optical pulses. *Lect Notes Phys* **545**, 100-122, (2000).
- 8 Kaindl, R. A. *et al.* Ultrafast mid-infrared response of $\text{YBa}_2\text{Cu}_3\text{O}_{7-d}$. *Science* **287**, 470-473, (2000).
- 9 Demsar, J., Podobnik, B., Evetts, J. E., Wagner, G. A. & Mihailovic, D. Evidence for crossover from a Bose-Einstein condensate to a BCS-like superconductor with doping in $\text{YBa}_2\text{Cu}_3\text{O}_{7-d}$ from quasiparticle relaxation dynamics experiments. *Europhys Lett* **45**, 381-386, (1999).
- 10 Gay, P., Stevens, C. J., Smith, D. C., Chen, C. & Ryan, J. F. Anisotropy of non-equilibrium quasi-particle dynamics in single-crystal $\text{YBa}_2\text{Cu}_3\text{O}_{7-d}$. *Physica C* **341**, 2269-2270, (2000).
- 11 Averitt, R. D. *et al.* Nonequilibrium superconductivity and quasiparticle dynamics in $\text{YBa}_2\text{Cu}_3\text{O}_{7-d}$. *Phys Rev B* **63**, 140502(R) (2001).
- 12 Luo, C. W. *et al.* Spatial symmetry of the superconducting gap of $\text{YBa}_2\text{Cu}_3\text{O}_{7-d}$ obtained from femtosecond spectroscopy. *Phys Rev B* **68**, 220508(R) (2003).

- 13 Semenov, A., Engel, A., Hubers, H. W., Il'in, K. & Siegel, M. Spectral cut-off in the efficiency of the resistive state formation caused by absorption of a single-photon in current-carrying superconducting nano-strips. *Eur Phys J B* **47**, 495-501, (2005).
- 14 Tinkham, M. *Introduction to superconductivity*. (McGraw-Hill, Inc., 1996).
- 15 Kozorezov, A. G. *et al.* Quasiparticle-phonon downconversion in nonequilibrium superconductors. *Phys Rev B* **61**, 11807-11819, (2000).
- 16 Semenov, A. D., Gol'tsman, G. N. & Korneev, A. A. Quantum detection by current carrying superconducting film. *Physica C* **351**, 349-356, (2001).
- 17 Prober, D. E. Superconducting Terahertz Mixer Using a Transition-Edge Microbolometer. *Appl Phys Lett* **62**, 2119-2121, (1993).
- 18 Segre, G. P. *et al.* Photoinduced changes of reflectivity in single crystals of $\text{YBa}_2\text{Cu}_3\text{O}_{6.5}$ (ortho II). *Physical Review Letters* **88**, 137001 (2002).
- 19 Rall, D. *et al.* Energy relaxation time in NbN and YBCO thin films under optical irradiation. *J. Phys.: Conf. Ser.* **234**, 042029 (2010).
- 20 Kabanov, V. V., Demsar, J., Podobnik, B. & Mihailovic, D. Quasiparticle relaxation dynamics in superconductors with different gap structures: Theory and experiments on $\text{YBa}_2\text{Cu}_3\text{O}_{7-d}$. *Phys Rev B* **59**, 1497-1506, (1999).